



# The influence of water percolation through crevasses on the thermal regime of Himalayan mountain glaciers

Adrien Gilbert[1], Anna Sinisalo[2], Tika R. Gurung[2], Koji Fujita[3], Sudan B. Maharjan[2], Tenzing C. Sherpa[2], and Takehiro Fukuda[4,5,a]

[1]Department of Geosciences, University of Oslo, Oslo, Norway

[2]International Centre for Integrated Mountain Development, GPO Box 3226, Kathmandu, Nepal

[3]Graduate School of Environmental Studies, Nagoya University, Nagoya 464-8601, Japan

[4]Graduate School of Environmental Science, Hokkaido University, Sapporo 060-0810, Japan

[5]Institute of Low Temperature Science, Hokkaido University, Sapporo 060-0819, Japan

[a]now at: Hokkaido Government, Sapporo 060-8588, Japan

*Correspondence to*: Adrien Gilbert (adri.gilbert3@gmail.com) and Anna Sinisalo (Anna.Sinisalo@icimod.org)

**Abstract.** In cold and arid climate, small glaciers with cold accumulation zone are often thought to be entirely cold based. However, Ground Penetrating Radar (GPR) measurements on Rikha Samba Glacier in the Nepal Himalaya reveal a large amount of temperate ice that seems to be influenced by the presence of crevassed areas. We used a coupled thermo-mechanical model forced by a firn model accounting for firn heating to interpret the observed thermal regime. We show that the addition of water percolation and refreezing in crevassed areas using a simple energy conservative approach is able to explain the observations. Model experiment shows that both steady and transient thermal regimes are significantly affected by latent heat release in crevassed areas. It makes half of the glacier base temperate, resulting in the dynamics mainly controlled by basal friction instead of ice deformation. Timescale of thermal regime change in response to warming climate is also greatly diminished with a potential switch from cold to temperate basal ice in 50-60 years in the upper part of the glacier while it would take 100-150 years without the crevasses effect. This study highlights the crucial role of water percolation through the crevasses on the thermal regime of glaciers and validates a simple method to take it into account in glacier thermo-mechanical models.



## 1. Introduction

The thermal regime of a mountain glacier controls its hydrology, flow rheology, and basal conditions affecting glacier dynamics and, consequently, its behavior in response to climate change. It influences erosion rates (Bennett and Glasser, 2009), potential glacier hazards (Faillettaz et al., 2011; Gilbert et al., 2015) and water resources in the glaciated catchments (Miller et al., 2012). It is thus essential to understand the processes causing and maintaining temperate basal conditions, as well as the mechanisms leading to changes in the thermal regime of glaciers.

Very little is known about thermal regime of the Himalayan glaciers due to the harsh conditions and logistical difficulties making direct observations challenging in the remote, high altitudes areas. Borehole temperature measurements, such as carried out on Khumbu, Yala and Gyabrag Glaciers in the Himalayas (Miles et al., 2018; Mae, 1976; Watanabe et al., 1984; Liu et al., 2009), provide direct observations of the glaciers thermal condition. However, a restricted number of boreholes give only very limited information about the spatial distribution of the ice temperatures within the glacier and need in that case to be extrapolate from numerical modeling to estimate the thermal structure of the glacier (Wang et al., 2018, Zhang et al., 2013).

Scattering of the electromagnetic signal in glacier ice is commonly interpreted as diagnostic of temperate ice in ground penetrating radar (GPR) data, and continuous GPR profiles can thus provide information about the spatial distribution of thermal ice zones within a glacier (e.g. Wilson et al., 2013; Gusmeroli et al., 2012; Irvine-Fynn et al., 2006; Pettersson et al., 2007). Wilson et al. (2013) showed that the interface between cold and temperate ice matching with the localization of temperatures reaching pressure melting point in the boreholes could be identified with a 10 MHz GPR on two sub-Arctic polythermal glaciers. In the Himalaya, such GPR data are rarer while Sugiyama et al. (2013) showed with GPR data that Yala Glacier in Nepal is polythermal, which was in agreement with previous two borehole measurements in the ablation and accumulation areas of the glacier (Watanabe et al., 1984).

In this study, we reveal the polythermal structure and ice thickness of a high altitude glacier in the Nepal Himalaya using GPR. We combine GPR data from 2010 and 2015 with other field data to determine ice thickness and to estimate the amount of temperate versus cold ice in the glacier. Measurements are interpreted using a 3D thermo-mechanical model for which we developed new methods to; (i) determine the thermal surface boundary condition, and (ii) take into account water percolation and refreezing in the crevassed areas. The model is forced by a surface mass balance model calibrated with the field measurements, and run to determine steady state and transient thermal regimes of the glacier. We compare our modeling results with the GPR data to conclude about processes defining the thermal regime of the glacier, and to provide recommendation on how to take them into account for further modeling studies.



## 2. Observations and data

### 2.1. Study area

Rikha Samba Glacier is a south-east orientated, medium sized glacier (5.5 km$^2$) located in the Hidden Valley in western
Nepal (Fig. 1a). The glacier is about 5.5 km long with an elevation ranging from 5420 to 6440 m a.s.l. in 2014. The glacier
melt water contributes to the Kali Gandaki Basin of Ganges River. The Hidden valley falls under a rain shadow climate
receiving the least precipitation in Nepal with an annual precipitation of 370 mm (Fujita and Nuimura, 2011). The annual
mean temperature measured with an automatic weather station (AWS) in vicinity of the glacier (Fig. 1a) was –5 °C in 2014.
The glacier was first visited in 1974, and it has been losing mass at a mean rate of -0.5 m w.e. yr$^{-1}$ between 1974 and 2010
(Fujita et al., 1997; Fujita and Nuimura, 2011).

### 2.2. Ground penetrating radar (GPR)

We used a Malå GeoScience ProEx ground penetrating radar (GPR) with a 30 MHz Rough Terrain Antenna (RTA) to
measure the ice thickness and thermal regime of Rikha Samba Glacier in 2015 (Fig. 1b). The continuous profiles were
filtered and some of them migrated, and the GPR reflectors were manually picked from the data. Picked two-way travel
times of the radar signal were converted to ice thickness using a wave velocity of $1.68 \times 10^8$ m s$^{-1}$ (Robin, 1975). Strong
scattering of the radar signal within the ice was interpreted as temperate ice whereas ice without internal reflectors was
classified as cold ice. In addition, we have also used point data collected in 2010 with an impulse GPR transmitter (Ohio
State University) with a set of half-wavelength 5 MHz dipole antennas (Fig. 1b). These data were used only for ice
thickness. The time increment of five years between the GPR measurements was corrected by projecting the 2010 data to
2015. This was done by assuming that the glacier was thinning with the same rate between 2010 and 2015 as the long-term
thinning rate for 100-m elevation bands obtained for a 12-year period between 1998 and 2010 (Fujita and Nuimura, 2011).

### 2.3. Glacier geometry and crevasse localization

A digital elevation model (DEM) was generated for Rikha Samba Glacier from Pleiades tri-stereo satellite imagery for
November 7, 2014. Crevassed areas on the glacier were visually identified from the imagery and refined using Google Earth
image (WorldView, September 21, 2012) in which the crevasses were more visible. The ice volume and bedrock topography
were initially estimated by defining ice thicknesses as zero at the margins of the glacier and interpolating the ice thickness
data measured with the GPR. For interpolation, we assumed a spherical semi-variogram and applied kriging algorithm. This
method is widely used to interpolate ice thicknesses measured with a GPR to estimate volumes of mountain glaciers (e.g.
Fischer, 2009). Since the GPR data do not cover the entire glacier, it results in high uncertainties in the interpolated bedrock
topography in some part of the glacier. The initial bedrock topography is thus corrected using the ice flow model (Sect.
3.4.2).



### 2.4. Glacier mass balance, surface velocities, and ice temperature

We constrained the surface mass balance model from stakes network measurements in 2012 and 2013 (Gurung et al., 2016) and from the total volume change estimated by geodetic survey (Fujita and Nuimura, 2011) over the period 1974-1994 (– 0.57 m w.e. a$^{-1}$) and 1998-2010 (–0.48 m w.e. a$^{-1}$). Stakes displacement monitored in the 1998-1999 period shows horizontal surface velocities between 9 and 24 m a$^{-1}$, which is greater than what could support the deformation of ice and thus revealing the existence of basal sliding (Fujita et al., 2001). Ice temperature at 10 m depth was measured with a thermistor chain in the lower ablation area (5600 m a.s.l., Fig. 1b) for 2014–2015. Air temperature and precipitation were observed with an AWS in vicinity of the glacier at 5320 m a.s.l. (Fig. 1a).

### 2.5. Meteorological parameters

We used the ERA-Interim reanalysis data (Dee et al., 2011) at daily timescale over the period 1980–2016 as an input data for the mass balance model. We only used air temperature data and assumed constant precipitation rate in time (no precipitation seasonality) to avoid complexity in the simulation. Temperature and precipitation are then distributed on the glacier according to altitudinal gradients to reproduce the observed mass balances. Bias in the ERA-Interim air temperature are calibrated using the local AWS data (Fig. 1b) over the period 2011-2015 by linear regression method.

### 3. Modeling methods

The modeling study aims to identify which physical processes lead to the thermal structure observed by the GPR measurements. First, we focus on steady state simulation for which, ice flow and thermal regime are in equilibrium with constant surface boundary conditions (surface temperature and mass balance). We then use the steady state simulation as initial condition of the transient model experiments.

### 3.1. Surface mass balance model

Mass balance is modelled using a degree-day method following Gilbert et al. (2016). Net annual surface mass balance is determined by:

$$B = C + R - M \tag{1}$$

where $B$ is the net annual surface mass balance (m w.e. a$^{-1}$), $C$ is the annual snow accumulation (m w. e. a$^{-1}$) , $R$ is the rate of refreezing (m w. e. a$^{-1}$) and $M$ is the annual melting (m w. e. a$^{-1}$).





In this study, we updated a degree-day model described in Gilbert et al. (2016) by including the influence of the spatial variability of the shortwave radiation to constrain both ice dynamics and thermal regime of the glacier. Meltwater is computed from the sum of two components (Pellicciotti et al., 2005):


$$m = \max\left[(T_a - T_{th}) \times f_m + S_{pot} \times f_{rad}; 0\right] \tag{2}$$

where $m$ is the daily melt (m w.e. d$^{-1}$), $T_a$ is the air temperature (K), $T_{th}$ is a temperature threshold for melting (K), $f_m$ is the melting factor (m w.e. K$^{-1}$ d$^{-1}$), $S_{pot}$ is the potential solar radiation (W m$^{-2}$) and $f_{rad}$ is the radiative melting factor (m w.e. W$^{-1}$ m$^2$ d$^{-1}$). Following a similar approach as used in Gilbert et al. (2016), $f_{rad}$ is computed from the radiative melting

factors for snow and ice ($f_{rad}^{snow}$ and $f_{rad}^{ice}$) and the ratio of the melting season during which the surface is snow covered ($r_{s/m}$):

$$f_{rad} = \begin{cases} f_{rad}^{snow} & if \ (r_{s/m} \geq 1) \\ f_{rad}^{ice} - (f_{rad}^{ice} - f_{rad}^{snow}) \times r_{s/m} & if \ (r_{s/m} < 1) \end{cases} \tag{3}$$

The annual ratio $r_{s/m}$ is computed assuming that:


$$r_{s/m} = \frac{C}{M} \tag{4}$$

The annual snow accumulation ($C$) and the annual amount of melting ($M$) are computed with $f_{rad}$ equal to $f_{rad}^{snow}$. Snow accumulation is calculated as a function of elevation ($z$, m a.s.l.):

$$C = \sum_{d=1}^{365} \begin{cases} P_{ref} \times \left(z - z_{p_{ref}}\right) \frac{dP}{dz} \frac{1}{100} & if \ T_a(d,z) < T_{snow} \\ 0 & if \ T_a(d,z) \geq T_{snow} \end{cases} \tag{5}$$


where, $P_{ref}$ is the daily precipitation rate (m w. e. a$^{-1}$) at the elevation $z_{p_{ref}}$ (m a.s.l.), $dP/dz$ is the precipitation lapse rate (% m$^{-1}$), $z$ is the elevation (m a.s.l.) and $T_{snow}$ is a temperature threshold that distinguishes between snow and rain (K). Assuming that refreezing in the previous year creates impermeable layers and thus occurs only to a depth equal to the annual accumulation rate, we write:


$$R = \min[M; f_r \times C] \tag{6}$$





where $f_r$ is a refreezing factor.

### 3.2. Thermo-mechanical model

The ice flow model is based on the Stokes equations for incompressible flow adopting Glen's flow law for viscous isotropic

ice (Cuffey and Paterson, 2010) and coupled to an energy conservation equation using the enthalpy formulation
(Aschwanden et al., 2012; Gilbert et al., 2014a):

$$\rho \left( \frac{\partial H}{\partial t} + \vec{v} \cdot \vec{\nabla} H \right) = \nabla (\kappa \vec{\nabla} H) \; + \; tr(\boldsymbol{\sigma}\dot{\boldsymbol{\epsilon}}) + Q_{lat} \tag{7}$$

where $\rho$ is the firn/ice density (kg m$^{-3}$), $t$ is the time (s), $H$ is the enthalpy (J kg$^{-1}$), $v$ is the ice velocity vector, $\kappa$ is the
enthalpy diffusivity (kg m$^{-1}$ s$^{-1}$), $Q_{lat}$ is the source term coming from meltwater refreezing (W m$^{-3}$), and $tr(\boldsymbol{\sigma}\dot{\boldsymbol{\epsilon}})$ is the strain
heating (W m$^{-3}$) with $\sigma$ and $\dot{\epsilon}$ respectively the stress and strain rate tensors. A basal heat flux of $4.0 \times 10^{-2}$ W m$^{-2}$ is assumed
for basal boundary condition. This value is not well constrain ranging from $2.0 \times 10^{-2}$ W m$^{-2}$ (observed at Rongbuk Glacier
in the Everest region (Zhang et al., 2013)) to $8.0 \times 10^{-2}$ W m$^{-2}$ as predicted by large scale model (Tao and Shen, 2008). The

enthalpy is defined from ice temperature $T_i$ (K) and water content ω:

$$H(T_i, \omega) = \begin{cases} \int_{T_0}^{T_i} C_p(T)dT & if \;\; T_i < T_m(p) \\[2mm] \int_{T_0}^{T_m(p)} C_p(T)dT \; + \; \omega L & if \; T_i = T_m(p) \end{cases} \tag{8}$$

where $C_p$ is the heat capacity of ice ($2.05\ 10^3$ J K$^{-1}$ kg$^{-1}$), $T_0$ is the reference temperature for enthalpy (set to 200 K), $T_m$ is
the melting point temperature (K), $L$ is the latent heat of fusion ($3.34\ 10^5$ J kg$^{-1}$) and $p$ is the pressure (Pa).

Changes in the glacier surface elevation are computed by solving a free surface equation (Gilbert et al., 2014a). The model is
solved using the finite-element software Elmer/Ice (Gagliardini et al., 2013) on a 3D mesh with a 50-m horizontal resolution
and 15 vertical layers. We adopt a linear friction law as a basal boundary condition for the Stokes equation:

$$\tau_b = \beta u_s \tag{9}$$

where $\tau_b$ is the basal shear stress (MPa), $u_s$ is the sliding velocity (m a$^{-1}$) and $\beta$ is the friction coefficient (MPa a m$^{-1}$).





### 3.3. Modeling crevasse influence via water percolation

In order to determine the areas where the crevasses are likely to form on the glacier, we compute the maximal principal Cauchy stress $\sigma_I$ (MPa) at the glacier surface from the stress tensor. We compare it with a threshold value $\sigma_{th}$ (MPa) to identify where damage production occurs (Pralong and Funk, 2005; Krug et al., 2014) and define the crevassed areas where
$\sigma_I > \sigma_{th}$.

In the crevassed areas, we make an assumption of free vertical percolation of the meltwater down to the bedrock, in which local surface meltwater is the only source of liquid water percolating into the crevasses. This means that any water coming from the surface runoff and draining to the crevassed area is neglected. Assuming that water refreezes in the first cold layer we compute a latent heat volumetric flux $Q_{lat}$ (see eq. 7) from the available annual meltwater and the ice temperature of the
current iteration. At each vertical layer i of the model, $Q_{lat}$ is computed from the amount of refreezing water $r_i$ (kg m$^{-2}$).

$$Q_{lat} = \frac{r_i}{dt \times dz_i \times L} \tag{10}$$

where $dz_i$ is the thickness of the layer $i$ (m) and $dt$ is the timestep (s). The amount of refreezing water $r_i$ is distributed from top to bottom with the condition that the enthalpy for a certain layer has first reached the value corresponding to the fusion of
water before the water access the next layer downwards. Starting from the surface melt $r_1 = m$, the amount of liquid water available for refreezing in the next layer $r_{i+1}$ is computed following:

$$r_{i+1} = \max\left[r_i - \frac{H_f - H}{L}; 0\right] \tag{11}$$

Using the estimated $Q_{lat}$ flux, a new steady state enthalpy field is computed and $Q_{lat}$ can be updated from the new
temperature field. The procedure is repeated until reaching a steady state. With this approach, the energy used to melt ice at the surface in crevassed area is released in the deeper ice body. It can be seen as an energy conservation approach rather than modeling of water routing through crevasses.

### 3.4. Strategy for steady state glacier

#### 3.4.1. Enthalpy surface boundary condition including firn/snow influence

For this study we develop a new method in order to determine surface boundary conditions for enthalpy. We use the 1D semi-parameterized approach developed in Gilbert et al. (2014b) and distribute it over the entire glacier. The method takes into account water percolation and refreezing in both firn and seasonal snow to determine the adequate surface boundary condition of the 3D model (Gilbert et al., 2012). The 1D model is solved on a one-dimensional 10-m-depth vertical profile at





each surface node of the 3D model. It allows a high temporal and vertical spatial resolution in order to explicitly solve
percolation and refreezing processes.

Starting from an initial uniform temperature profile, firn/ice temperature is solved at daily time steps along the vertical
profile with a 0.06 m resolution. The 1D model is forced by air temperature and by the surface mass balance model that
provides snow accumulation and surface melting for the corresponding surface node. To compute steady state condition, the
model is driven by a mean annual cycle of air temperature which is determined at daily time scale from the meteorological
data. A Gaussian random noise is added to the computed mean annual temperature cycle to plausibly represent the daily
temperature variability. The standard deviation of the Gaussian function is adjusted to match the number of positive degree-
days in our mean annual cycle to the mean one in the data. The 1D model run during several years with the same cycle until
the 10m-depth temperature (approximate limit of the thermally active layer) reach a mean annual equilibrium value that will
be used as boundary condition of the thermo-mechanical model.

At each surface node, the initial density profile is calculated from the steady state firn thickness $F_{ref}$ (m w.e.) which is
computed from the steady state mass balance assuming:

$$F_{ref} = \begin{cases} \dfrac{B}{t_{yr} \times d_f} & \text{if } B \geq 0 \\ 0 & \text{if } B < 0 \end{cases}$$

(12)

where $d_f$ is a firn densification rate parameter (s$^{-1}$) and $t_{yr}$ (s) is one year in second. The density is then calculated assuming
a linear evolution of density with depth between the surface density $\rho_0$ (kg m$^{-3}$) and the ice density $\rho_{ice}$ (kg m$^{-3}$) at the
firn/ice interface. It gives:

$$\rho(z) = \rho_0 + \frac{(\rho_{ice} - \rho_0)}{(z_s - z_{ice})}(z_s - z)$$

(13)

where z is the vertical coordinate (m a.s.l.), $z_s$ is the elevation of the surface (m a.s.l.), and $z_{ice}$ is the elevation of the firn/ice
transition (m a.s.l.). From mass conservation, $z_{ice}$ has to satisfy:

$$\int_{z_{ice}}^{z} \left( \rho_0 + \frac{(\rho_{ice} - \rho_0)}{(z_s - z_{ice})}(z_s - z) \right) dz_f = \rho_w F$$

(14)

where $\rho_w$ is the water density (1000 kg m$^{-3}$), and $F$ is the firn thickness (m). Combining the equation 13 and 14 gives:



$$\rho(z_f) = \min\left[\rho_0 + \frac{(\rho_{ice}^2 - \rho_0^2)}{2\rho_w F}(z_s - z); \; \rho_{ice}\right] \tag{15}$$


In order to take into account the snow seasonal variability due to snow/rain threshold (Table 1), the density profile is updated at each time step by computing the evolution of $F$:

$$F(t + dt) = \max\left[F(t) + (c(t) - d_f F) \times dt; 0\right] \tag{16}$$

where $c(t)$ is the daily net surface accumulation (m w.e. d$^{-1}$). The density profile and the surface elevation are updated only if $F > F_{ref}$ by adding the corresponding amount of snow at density $\rho_0$. The initial value of $F$ is set to $F_{ref}$.

### 3.4.2. Bedrock Topography and basal sliding condition

The main challenge in determining the glacier thermo-mechanical equilibrium is that: (i) the bedrock topography is not resolved everywhere underneath the glacier, and (ii) the glacier is sliding, which means that a friction coefficient has to be
quantified. In order to resolve these issues, we used the following approach.

Step 1: Starting from the measured surface topography, we run the coupled thermo-mechanical model with the interpolated bedrock topography and a uniform basal friction coefficient during a 10-year period forced by the steady state surface mass balance and enthalpy. In order to obtain a steady state mass balance, we shifted our temperature forcing to obtain balanced mean conditions during the simulation period. Here, we assume a friction coefficient ($\beta$) that allows the best match with the
measured surface horizontal velocities ($10^{-2}$ MPa a m$^{-1}$).

Step 2: The computed changes in the free surface in Step 1 are reported to the bedrock topography.

Step 3: After a few iterations between Steps 1 and 2, we obtain a corrected bedrock topography where major flux divergences are avoided. Using this bedrock topography and the measured surface topography we invert for the friction coefficient $\beta$ by constraining the surface velocity on emergence velocities, which are taken opposite to surface mass balance.
This is done by using a controlled inverse method to minimize a cost function defined from the misfit with measured surface velocities and a regularization term (Gillet-Chaulet et al., 2012; Gagliardini et al., 2013). Following Gilbert et al. (2016, 2018) we define the cost function from the misfit between modelled and measured emergence velocities.

Step 4: We finally run the model using corrected bedrock topography and inverted friction coefficient until the surface topography reaches a steady state.

This method allows reaching a thermo-mechanical equilibrium in which surface topography and velocities are in reasonable accordance with the observation which allows a realistic study of the glacier thermal regime.



### 3.5. Transient evolution

Transient simulations are performed at yearly timestep using the steady state glacier as initial condition. Surface mass
balance and enthalpy are updated each year from the surface model described previously and forced by daily temperature
reanalysis. We assume constant basal friction parameter through time.

## 4. Results

### 4.1. Thermal regime and ice thickness measured with the GPR

GPR data show that Rikha Samba Glacier is a polythermal glacier consisting mainly of cold ice (Figs. 2 and 3). The
measured maximum thickness was 178 ± 2 m in the middle part of the glacier where the surface slope is relatively gentle. In
contrast to the ice of Yala Glacier, another polythermal glacier in the Nepal Himalaya (Sugiyama et al., 2013), temperate ice
is also found in the ablation area of Rikha Samba Glacier and only the lowermost part of the glacier where ice thickness is
less than 25 m is completely cold in this area. Ice temperature measurements by the thermistor chain support the GPR
interpretation of an upper cold ice layer with sub-zero temperatures at the depth of 10 m (annual mean –2 °C) in the ablation
area of the glacier (Fig. 1). A notable characteristic of the GPR based thermal regime is that temperate ice localization seems
to be associated with the presence of surface crevasses (Figs. 2 and 3).

### 4.2. Surface mass balance

We run the mass balance model using the 2014 surface topography over the period 1980-2016 using calibrated temperature
reanalysis data (ERA-Interim) and assuming a constant precipitation rate. The parameters were constrained by the stake
measurements in 2012/2013 (Gurung et al., 2016), meteorological observations (Gurung et al., 2016) and geodetic mass
balance over the periods 1974-1994 and 1998-2010 (Fujita and Nuimura, 2011) (Fig. 4). The parameters are summarized in
Table 2.

Balanced conditions for the 2014 geometry are reached for a climate that is 0.7°C colder than the 1980-2016 climate with an
Equilibrium Line Altitude (ELA) of 5770 m a.s.l. (1980-2016 ELA is 5880 m a.s.l.; Fujita and Nuimura, 2011). This
provides a mean surface mass balance and a melting rate to force the steady state glacier simulation (Fig. 4a).

The model provides a good agreement with the observations but is not able to reproduce the same mass balance distribution
as observed in 2013 (Figs. 4b-d). Interannual variability of the mass balance produced with our mass balance model is
probably not very well represented since we assume a constant precipitation rate. Furthermore, Rikha Samba Glacier is a
summer accumulation type of glacier in which precipitation events in summer can significantly affect the mass balance
through albedo feedback (Fujita and Ageta, 2000; Fujita, 2008). However, the long-term trend and the mass balance gradient
agree with the observations, which is satisfactory for the purpose of this study focusing on the thermal regime.





### 4.3. Enthalpy surface boundary condition

The modeled upper boundary condition field (Fig. 5) revealed mainly cold surface condition with a temperate band between 5800 and 5900 m a.s.l. where both melting and firn thickness are sufficient to maintain temperate conditions. In the higher

part of the accumulation area, the water percolation occurs only in the first two meters due to limited amount of meltwater resulting in cold temperature at 10 m-depth (see Site 1 in Fig. 5), whereas lower down at Site 2 meltwater percolates deep enough to keep temperate conditions all year round at 10 m-depth. In the ablation zone (Site 3), water percolation is limited to the seasonal snow thickness resulting in cold boundary condition.

### 4.4. Modeled steady state glacier

The modeled steady state glacier is in good accordance with measured ice thickness (Fig. 6), measured horizontal velocities (Fig. 7a), and observed crevassed areas (Fig. 7c). The correction made on the bedrock topography following the method described in section 3.4.2 greatly improved the quality of the modeling in the parts where radar measurements are inexistent (Fig. 6). A simple interpolation (Fig. 6a) leads to non-physical ice thickness with unrealistic flux divergence, which are avoided by our method. The inversion of the basal friction coefficient (Fig. 7b) provides a final steady state where ice flow is

in accordance with steady state emergence velocities. The good agreement with horizontal velocities measured at stakes (Fig 7a) shows that our estimated emergence velocities (from surface mass balance) are consistent with the observed ice flow.

### 4.5. Thermal regime: influence of melt water percolation

Modeling results show that water percolation in crevasses strongly affects the steady state thermal structure of the Rikha Samba Glacier leading to large temperate zones even at the glacier bed (Figs. 8b and 9b). It significantly extends the

temperate based parts, which cover almost the entire ablation area. Although we adopted a simple approach for water percolation through crevasses, modeled temperate ice thickness is in fairly good agreement with the GPR data (Fig. 8b). If water percolation through crevasses is neglected, the thermal regime of the glacier forced by mainly cold upper boundary conditions (Fig. 5) would result in a mainly cold based glacier (Fig. 9a). In this case, cold ice advection from the higher part of the glacier is able to compensate for the temperate surface conditions of the lower accumulation zone (Fig. 5), and only

two bands of temperate ice are able to reach the bed on both sides of the flow line of the glacier (Fig. 9a). Such thermal regime is in large disagreement with observed amount of temperate ice from the GPR data (Fig. 8a). This indicates a significant role of deep water percolation through cracks in cold ice as suggested by the GPR observations. We show that the use of observed (8b and 9b) or modeled (8c and 9c) crevassed area lead to similar result and validate our approach to model the localization of crevasses.



### 4.6. Transient evolution

Despite the good agreement between the GPR data and the steady state model, a significant difference exists at the highest crevasse field. A temperate area is clearly visible in the GPR data (Fig. 3a) whereas steady state thermal regime model predicts cold ice (Fig. 8). Mass balance measurements show that Rikha Samba Glacier has not been at equilibrium state for at least 40 years with an almost constantly negative rate of –0.5 m w.e. a$^{-1}$ (Fig. 4c). This temperate area could be therefore the signature of a transient response to the climate change. In order to investigate the potential impact of 40 years of unbalanced state on the glacier thermal regime, we performed a transient simulation starting in 1975 from the modeled steady state (experiment with observed crevasses) and forced by the reanalysis time-series. This shows that upper boundary conditions changed significantly with a cooling of the former accumulation zone in response to firn disappearance and a warming of the highest elevation due to melting increase (Fig. 10b). After a 40-years run, a temperate zone that did not exist at the steady state, developed in the highest crevassed area (Figs. 10c-d) as observed on the 2015 GPR measurements (Fig. 10d). This result strongly suggests that the presence of temperate ice in this zone is a result of a transient response to the climate change and increasing surface melting at the higher elevations. Results also agree better with the observations, including the thermistor data (Figs. 10a).

To assess the sensitivity of the thermal regime to future temperature change, we performed a future simulation of the glacier retreat and thermal change until 2100 with (Fig. 11) and without (Fig. 12) water percolation through crevasse for a linear temperature trend increase of +1 °C between 2014 and 2100 (+1.7 °C in comparison with the steady state climatic condition). This shows a much faster development of a new temperate area when water percolation in the crevasses is taken into account. In this case, the glacier becomes almost entirely temperate based by 2050 (Fig. 11) whereas it would remain almost entirely cold if water percolation through the crevasses would be neglected (Fig. 12). This highlights the crucial role of deep water percolation through cracks in the thermal regime of the polythermal glaciers. A phenomenon that should be taken into account together with firn heating when modeling past and future responses of thermal regimes and retreat of glaciers.

## 5. Discussion

### 5.1. Uncertainty

The modeled thermal regime is sensitive to the basal heat flux and the firn thickness, which are poorly constrained. Sensitivity experiment on those two parameters for steady state simulation shows that the amount of modeled basal temperate ice can vary significantly but the thickness of the modeled temperate ice always remains much less than indicated by the GPR observations if the crevasse influence is not taken into account. This means that uncertainty on those parameters cannot explain the disagreement between data and model when the role of crevasses is neglected. The mass balance model we used is simplified since seasonality and time variation in precipitation are not taken into account. However, the purpose





of the study is not an accurate simulation of Rikha Samba glacier past and future evolution, but a study about its thermal regime. Our study relies on long-term mean value of surface mass balance. This should be adequately captured by our simple model which is calibrated on data.

Since density of ice is well constrained and there was no snow or firn on the glacier at the time of the field measurement in
2015, the main uncertainties of the GPR measurements arise from the GPS positioning of the GPR measurements, the radar wavelength and scattering of the radar signal. For the point measurements and those parts of the GPR profiles along which the bedrock reflection was clearly identified, the accuracy of the horizontal coordinates is about ±20 m especially on the steepest surface slopes. In addition, vertical resolution of the GPR signal is usually considered to be approximately one quarter of the radar signal wavelength, which is about 5.6 m and 33.6 m for the 30 MHz and 5 MHz antennas, respectively.
In other words, the vertical resolution of the englacial scattering interpreted as temperate ice and ice thickness along the continuous GPR profiles is about 1.4 m, whereas the same for the ice thickness obtained from the point measurements is about 8.4 m. In addition, limited coverage of the radar profiles on the glacier introduce uncertainty in the bedrock topography inferred from the GPR data even after correction using the model. Our interpretation of the thermal regime based on the englacial radar scattering of the GPR 30 MHz profiles is supported by previously found close agreement between the
observed scattering and borehole temperatures without significant difference in observed englacial scattering relative to the expected measurement error at 10, 35 and 50 MHz antenna frequencies (Wilson et al., 2013).

In the modeled bedrock topography, the difficulty arises from the fact that friction coefficient is unknown and we had to assume a uniform value of basal friction coefficient to correct the bedrock topography from flux divergence anomalies. The friction coefficient we inferred in a second step to force the steady emergence velocity to match the balanced surface mass
balance is therefore affected by ice thickness errors. The resulting velocity field is consistent with mass flux conservation but contains uncertainty in the respective contribution of the basal sliding and ice deformation. It makes delicate to interpret the modeled basal friction (Fig. 7b), which has to be seen more as a tuning parameter rather than a parameter revealing physical processes. However, these uncertainties have little influence on the modeled thermal regime since advective processes will be still correctly represented as long as the surface velocities match with the observations (Fig. 7a).

**5.2.  Role of surface runoff**

Our study shows that the thermal regime of Rikha Samba Glacier can be modeled by taking into account melting occurring in crevassed fields and neglecting water input coming from the surface runoff. For a polythermal glacier with a cold surface layer, which is a common feature in the ablation areas where firn heating is nonexistent, surface runoff occurs in well-marked and persistent streams at the surface (Ryser et al., 2013). Those streams bring water into the crevasse fields through
few localized entry points only. Thus, there is a relatively small contact surface between the streams and cold ice, and only a limited amount of water will refreeze (Fujita et al., 1996). This is why the influence of surface runoff on the thermal regime of the glacier is likely to remain limited. Similarly, Lüthi et al. (2015) conclude that moulins have little influence on the thermal regime of Greenland Ice Sheet since they can be represented as line sources that provide limited warming of the





surrounding ice. In contrary, surface melting occurring on the crevasses field is well distributed and can release latent heat
on much larger areas, having a stronger impact on thermal regime.

### 5.3. Enhanced influence of climate change on glacier thermal regime and dynamics

The influence of deep latent heat release through melt water percolation in crevasses have been already observed in Greenland. Lüthi et al. (2015) observed temperature anomalies in borehole measurements that can only be explained by latent heat released down to 400 m-depth in the crevasse fields. Similar conclusions have been made by Hills et al. (2017)
although they show that these effects remain localized and may not really influence the thermal regime of the Greenland Ice Sheet at large scale. In the case of mountain valley glaciers, crevasse fields can cover a significant fraction of the total glacier area. This is combined with a generally much faster ice flow leading to efficient advective processes that transport the heat produced in the crevassed areas. The results of these combined effects are significant and greatly influence the thermal regime at the glacier scale as shown in this study. As already pointed out for ice sheets (Phillips et al. 2010, 2013), the
timescale of the glacier thermal regime response to climate change is also greatly diminished compared to the case where only heat diffusion/advection of surface changes are taken into account. However, we show here that deep water percolation is likely restricted to the crevassed areas and absent elsewhere in order to reproduce the observed thermal structure. This restricts the spatial extent of the process on the glacier as it is dependent on the bedrock topography and related crevasse localization. Nevertheless, it is likely that the meltwater percolation via crevasses has a significant impact on the thermal
regime as highlighted here with the Rikha Samba Glacier, and that it is a common for all polythermal high altitude glaciers. Future climate change could also lead to faster thermal regime response than previously thought (Gilbert et al. 2015), especially in the cold accumulation areas where the pre-industrial melting rates were not sufficient to form temperate ice.

### 6. Conclusion

In this study, we use GPR measurements to show that the high elevation Himalayan Rikha Samba Glacier is polythermal.
We interpret the field observations of the temperate ice thickness using a 3D thermo-mechanical model constrained by a surface model taking into account water percolation in firn and seasonal snow. We show that the firn/snow heating, heat deformation and geothermal heat flux alone cannot explain the observed amount of temperate ice. The combining evidence of model and observations reveal that valley-type mountain glaciers in cold climate are greatly affected by water percolation into crevassed fields releasing latent heat into the ice body. It affects the thermal regime at the scale of the whole glacier
making temperate ice zones much larger than they would be without this effect. This allows sliding in large areas of the bed and largely affect the glacier dynamics and ice thickness.

We also show that thermal regime of Rikha Samba Glacier is affected by a transient response to the last 40 years climate change extending the temperate area to the highest part of the glacier. The thermal changes are occurring at a much smaller timescale (~50 years) due to the crevasse effect compared to what it would be by advection/diffusion processes only (>100



years). Our study reveals the crucial role of deep water percolation through cracks in determining both steady state and transient thermal structure of the polythermal glacier. We provide a simple approach easily applicable to any glacier for a more accurate reconstruction of complex thermal structures as observed on Rikha Samba Glacier.


*Data availability*. As required by ICIMOD publication policy, GPR data used in this study will be made publicly available through ICIMOD's Regional Database System at: http://rds.icimod.org. ICIMOD has an open data policy.

*Author contributions*. AS and AG designed the study. AG performed the modeling analysis. AS, TRG, TCS conducted the
field work in 2015. KF and TF conducted the fieldwork in 2010. AS and TF analyzed the GPR data. TRG and KF analyzed the ERA-Interim data and mass balance. SBM analyzed the DEM data. AG and AS wrote the paper with contributions from all other authors.

*Competing interests*. The authors declare that they have no conflict of interest.

*Acknowledgements*. This work was supported by ICIMOD's Cryosphere Initiative, funded by Norway and by ICIMOD core funds contributed by the Governments of Afghanistan, Australia, Austria, Bangladesh, Bhutan, China, India, Myanmar, Nepal, Norway, Pakistan, Sweden, and Switzerland. The authors would like to thank L. Rai for his support in the field measurements in 2015 and J. Gustafsson and T. Zwinger their inputs when designing study as well as the entire field teams
in 2010 and 2015. Adrien Gilbert acknowledges the Univ. Oslo EarthFlows initiative and funding from the European Research Council under the European Union's Seventh Framework Programme (FP/2007-2013)/ERC grant agreement no. 320816. The views and interpretations in this publication are those of the authors' and they are not necessarily attributable to their organizations.




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




**Table 1. Parameters of the surface model for enthalpy boundary condition.**

| Name | Symbol | Values | Units |
|------|--------|--------|-------|
| Residual water saturation [a] | $S_r$ | $5.00 \times 10^{-3}$ | |
| Thermal conductivity of snow [a] | $k$ | $f(\rho)$ [b] | W K$^{-1}$ m$^{-1}$ |
| Firn densification rate | $d_f$ | $8.20 \times 10^{-2}$ | d$^{-1}$ |
| Surface density | $\rho_0$ | 350 | kg m$^{-1}$ |
| Ice density | $\rho_{ice}$ | 917 | kg m$^{-1}$ |
| Latent heat of fusion | $L$ | $3.34 \times 10^5$ | J kg$^{-1}$ |
| Heat capacity of ice | $C_p$ | 2050 | J kg$^{-1}$ K$^{-1}$ |

[a] Described in Gilbert et al. (2014b); [b] Formulation proposed by Calonne et al. (2011)





**Table 2. Parameters of the surface mass balance model.**

| Name | Symbol | Values | Units |
|---|---|---|---|
| Melting factor | $f_m$ | $1.2 \times 10^{-2}$ [b] | m w.e. d$^{-1}$ K$^{-1}$ |
| Radiative melting factor for ice | $f_{rad}^{ice}$ | $9.3 \times 10^{-5}$ [b] | m w.e. d$^{-1}$ W$^{-1}$ m$^2$ |
| Radiative melting factor for snow | $f_{rad}^{snow}$ | $4.6 \times 10^{-5}$ [b] | m w.e. d$^{-1}$ W$^{-1}$ m$^2$ |
| Precipitation Lapse Rate | $dP/dz$ | 43 [a] | % km$^{-1}$ |
| Annual precipitation | $P_{ref}$ | 374 [c] | mm a$^{-1}$ |
| Reference elevation for $P_{ref}$ | $z_{ref}$ | 5310 | m a.s.l. |
| Temperature Lapse Rate | $dT/dz$ | $-6.2 \times 10^{-3}$ [a] | K m$^{-1}$ |
| Snow/rain threshold | $T_{snow}$ | 276.75 [a] | K |
| Melting threshold | $T_{th}$ | 272.65 [b] | K |
| Refreezing factor [d] | $f_r$ | 0.15 [b] | – |

[a] Gurung et al. (2016); [b] Calibrated in this study; [c] Fujita and Nuimura (2011); [d] Gilbert et al. (2016)





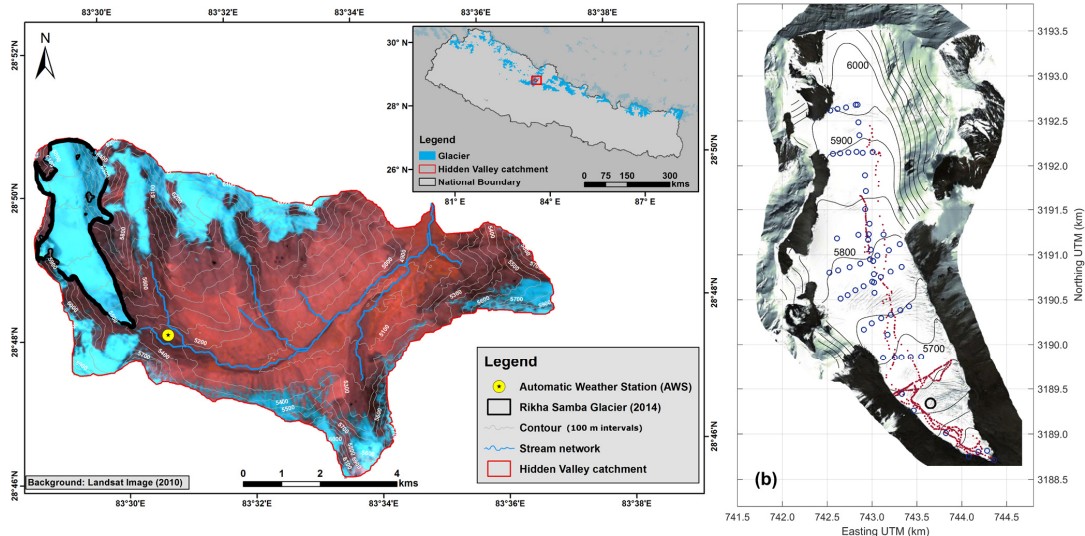


**Figure 1. (a) Location and (b) map of Rikha Samba Glacier in the Hidden Valley catchment in Nepal. Radar tracks in 2015 (red dots), radar point measurements in 2010 (blue circles), and location a thermistor chain (black circle) are shown in (b). Background image of (a) is of Landsat 5 in 25 May 2010.**


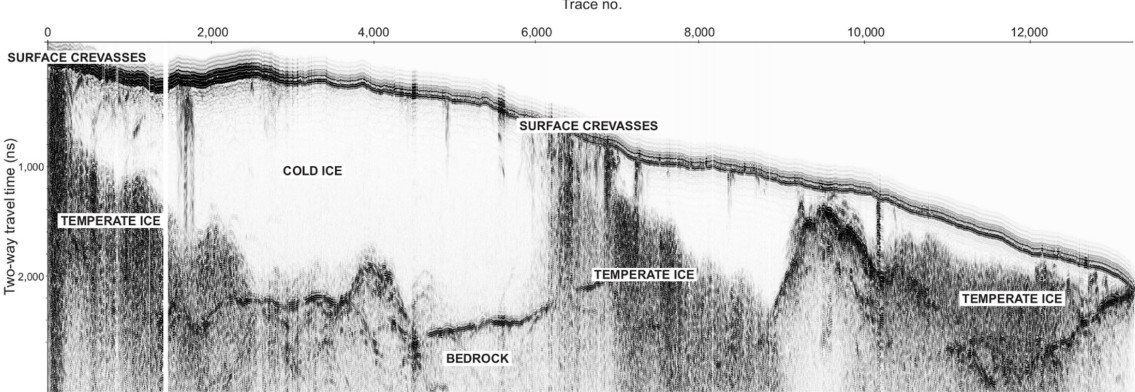

**Figure 2. 30 MHz radar profile measured in 2015 along the black dashed line in Fig. 3a.**





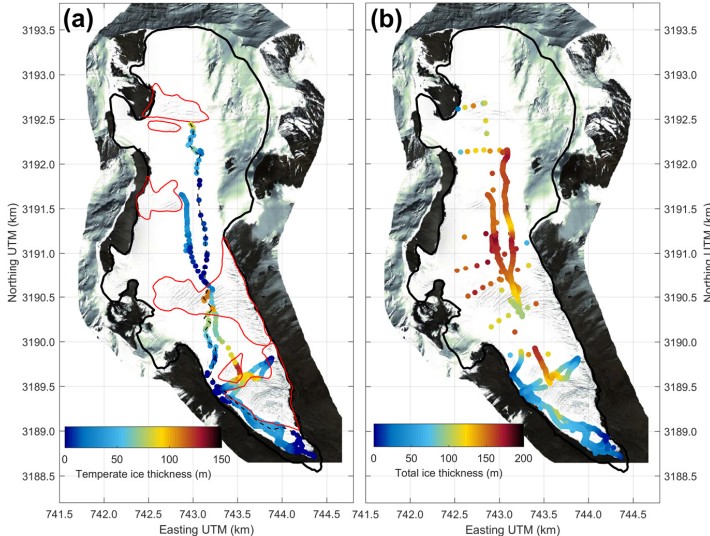

Figure 3. (a) Measured temperate ice thickness (dots) and observed crevassed areas (red lines). (b) Measured total ice thickness.






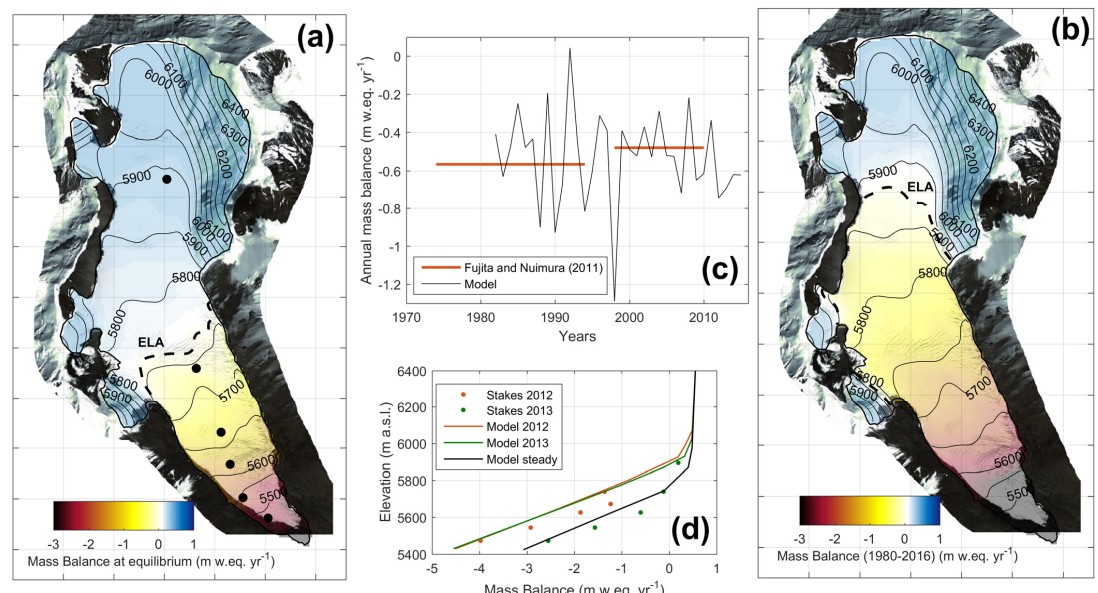

**Figure 4. Modeled mass balance using ERA-interim reanalysis data. (a) Surface mass balance at equilibrium. Black dots are stakes localization from Gurung et al. (2016). (b) Mean surface mass balance during the period 1980-2016. (c) Modeled annual surface mass balance compared to geodetic data (Fujita and Nuimura, 2011). (d) Modeled surface mass balance as a function of elevation at equilibrium (black line), in 2012 (red line) and 2013 (green line). Dots are stakes measurements for the years 2012 and 2013 (Gurung et al., 2016).**




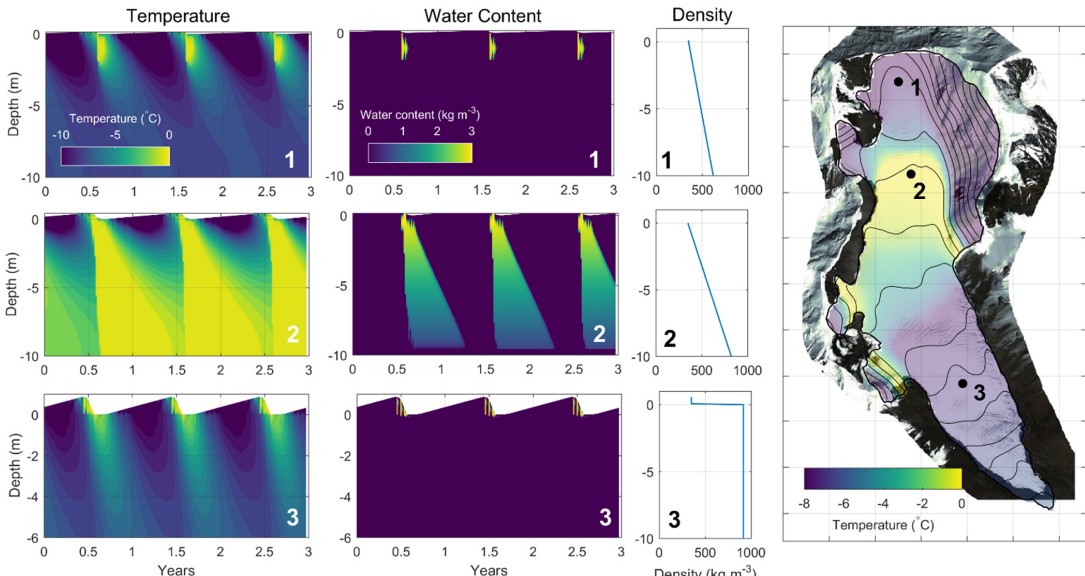

**Figure 5. One-dimensional temperature, water content and density modeled in the first 10m-depth during 3 years and at three different localizations on the glacier. The one-dimensional model is forced by a reference temperature/precipitation annual cycle until reaching steady state condition at 10 m-depth. Mean annual temperature at 10 m-depth is then used as an upper boundary condition for the thermo-mechanical model (mapped on the right panel).**

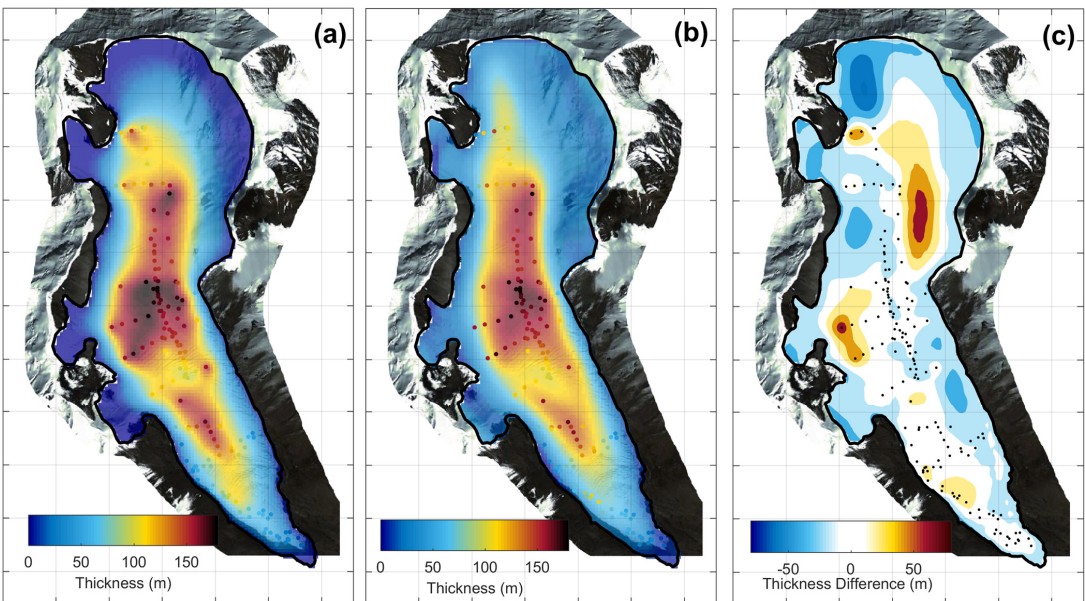

**Figure 6. (a) Interpolated ice thickness from GPR measurements (dots in the three panels). (b) Ice thickness after bedrock topography smoothing and correction using free surface relaxation. (c) Difference between interpolated and corrected ice thickness and localization of the GPR measurements (black dots).**





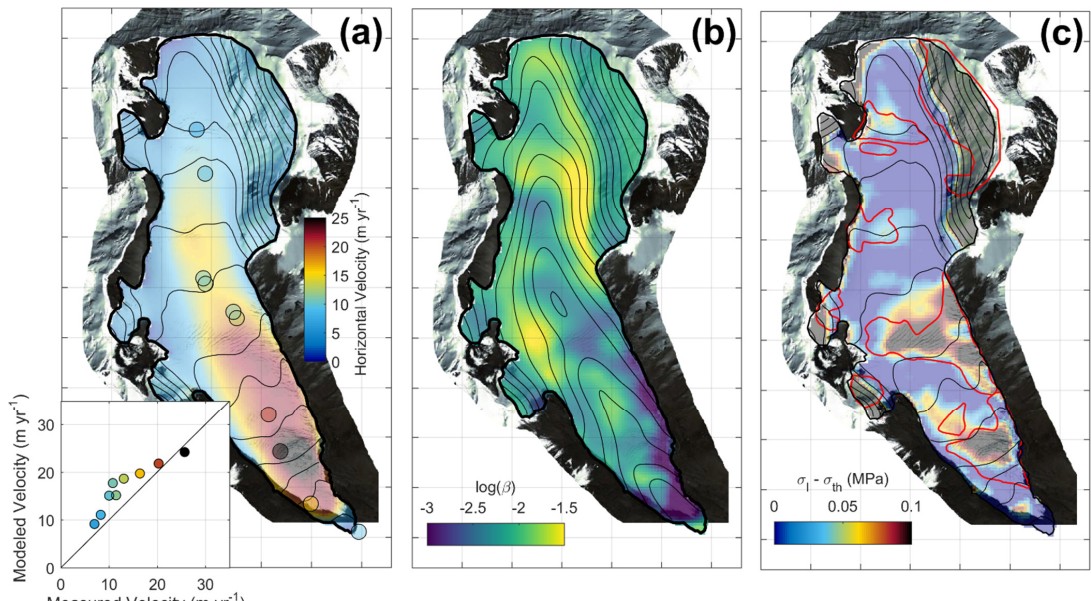

**Figure 7. (a) Steady state surface velocities compared with the measurements (colored dots and inset). Contour lines are surface topography with 50 m intervals (b) Friction coefficient inferred from emergence velocities (assumed to be opposite of surface mass balance). Contour lines are bedrock topography with 50 m intervals (c) Modeled crevasse localization from maximal Cauchy stress anomaly (color scale) compared with observations (red lines).**





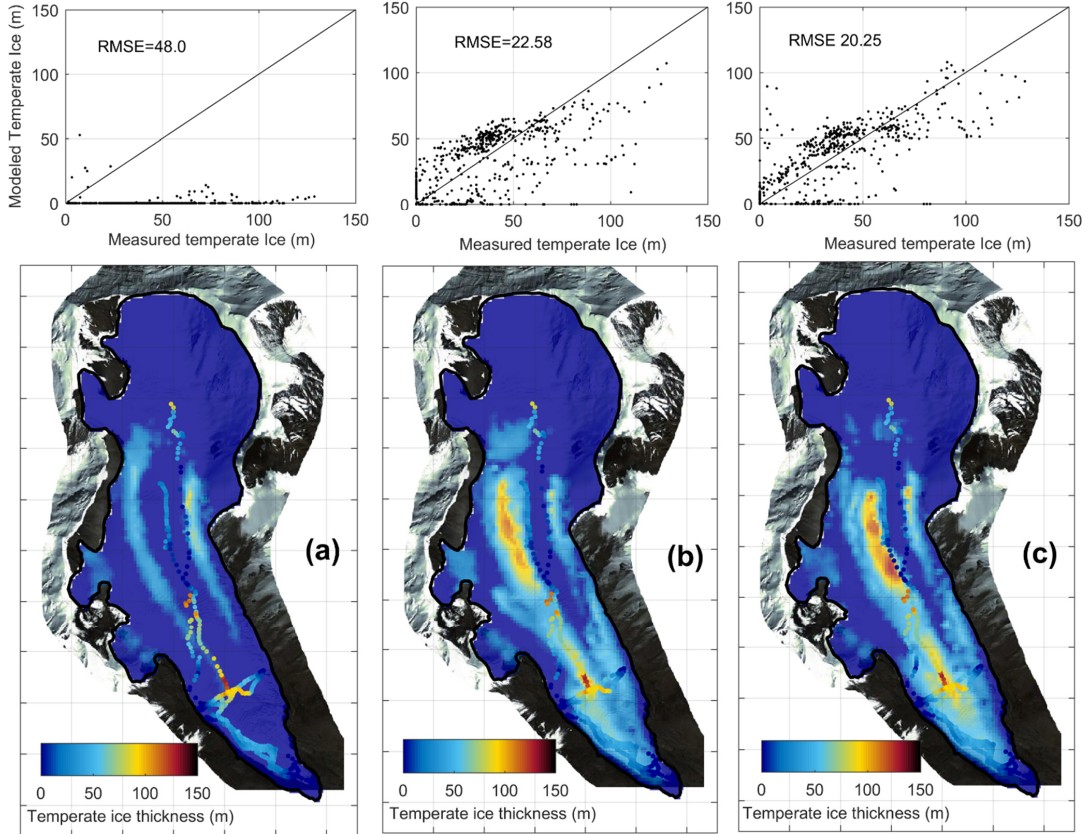

**Figure 8. Modeled vs. measured temperate ice thicknesses (upper panel) and comparison between model (color background) and measurements (dots) (lower panel): (a) without crevasse influence, (b) With water percolation in observed crevassed areas, and (c) with water percolation in modeled crevassed areas.**




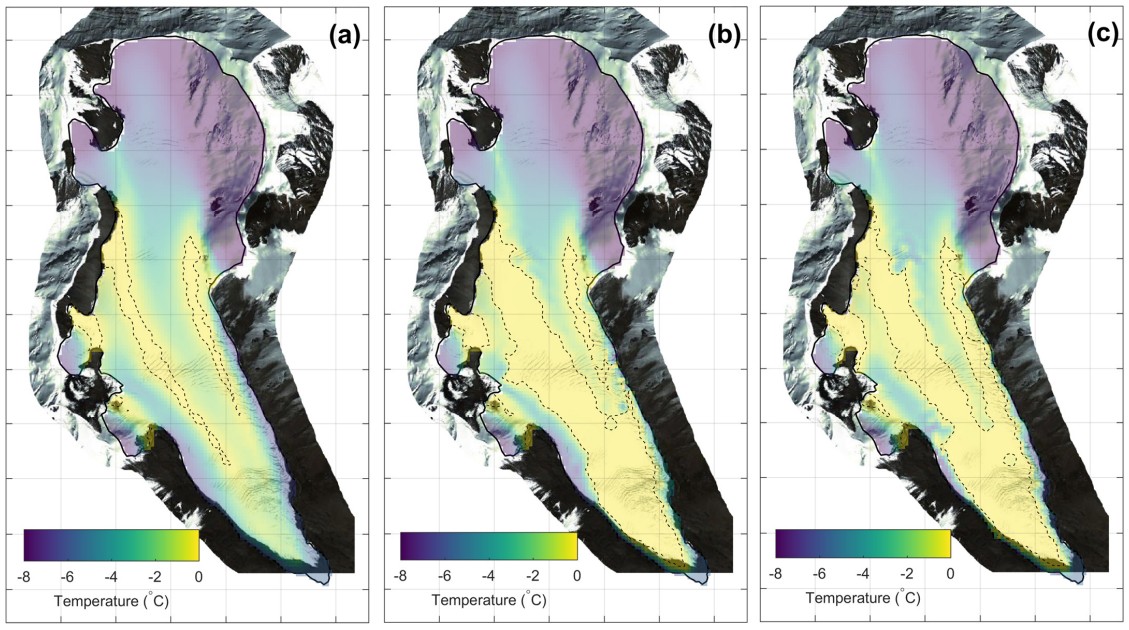

**Figure 9. Modeled steady state basal temperature. (a) Without crevasse influence. (b) With water percolation in observed crevassed areas. (c) With water percolation in modeled crevassed areas. Dashed lines delimit the temperate areas.**

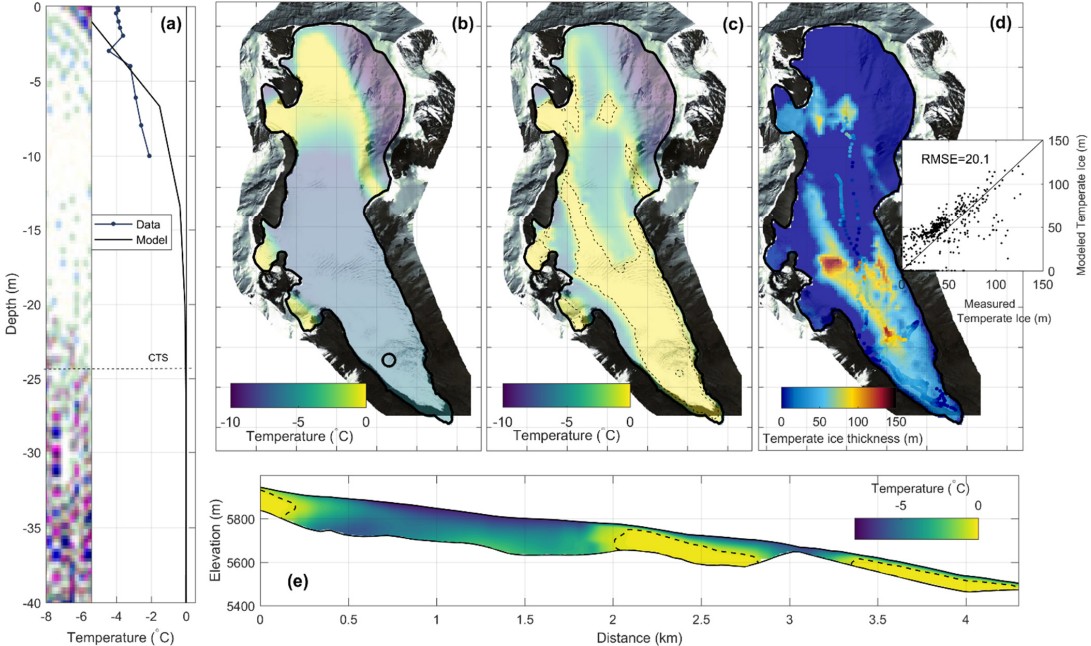

**Figure 10. Modeled thermal regime after a 40-years transient run forced by mean climatic condition over the period 1981-2016 (see Fig. 3b). (a) Modeled and measured mean temperature profile (2014/2015) in ablation zone (black circle in (b)). Inset is the**
**radar section next to the thermistor (black line in (e)) with the dashed line showing the modeled CTS. (b) Mean surface boundary condition over the period 1981-2016. (c) Modeled basal temperature. (d) Distribution of modeled and measured temperate ice thickness. (e) Modeled temperature along the radar cross section presented in Fig. 2.**
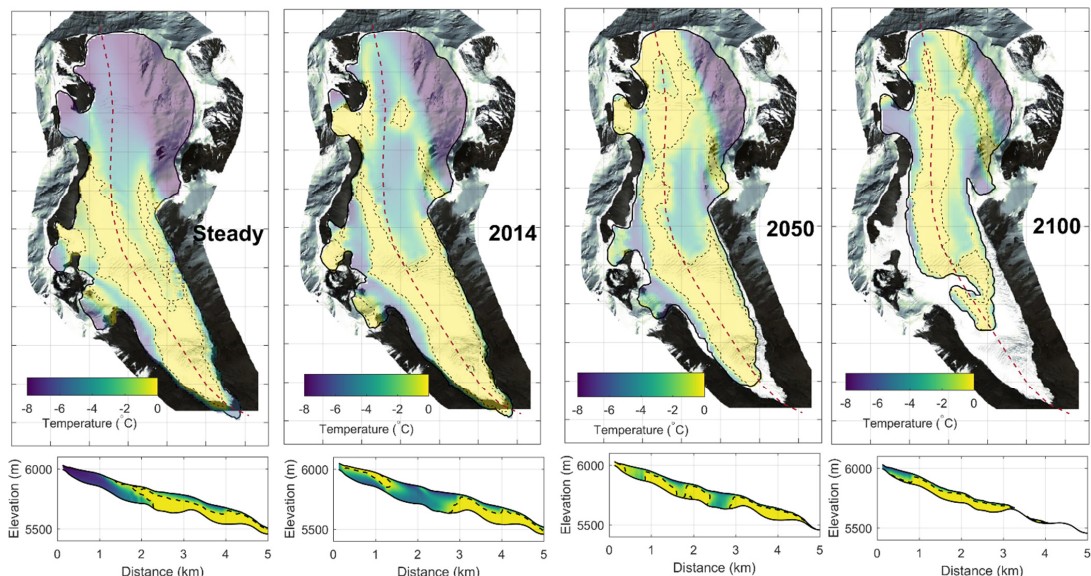

**Figure 11. Future evolution of Rikha Samba Glacier assuming linear temperature increase of +1 °C between 2014 and 2100 (+1.7**
**°C in comparison with the steady state climatic condition). Upper panels represent basal temperature evolution. Lower panels are**
**temperature evolution along the middle cross section.**





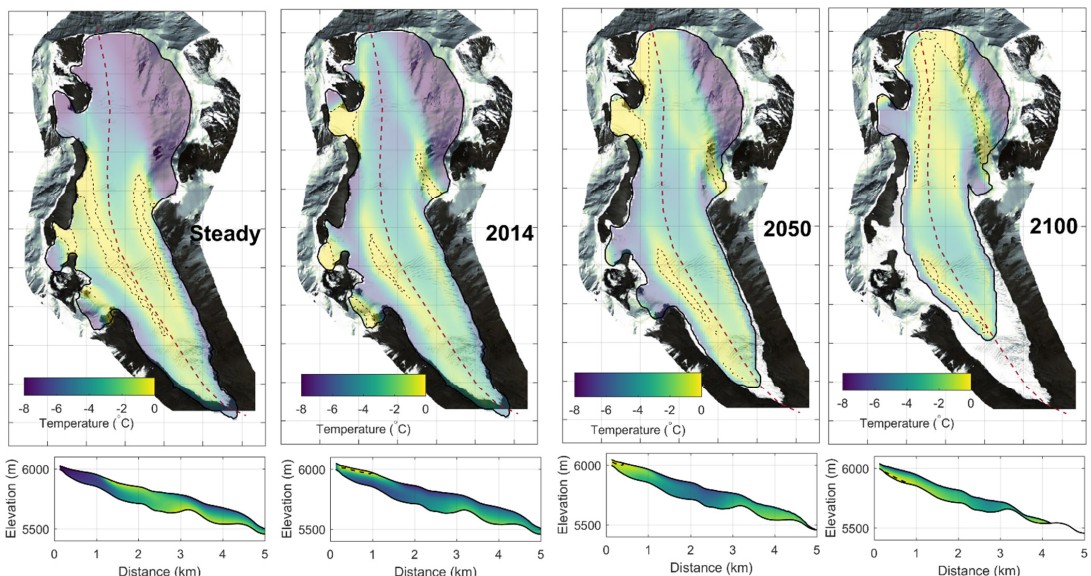

**Figure 12. Same as Fig. 11 but without water percolation in crevasses.**