# Peer review of "The influence of water percolation through crevasses on the thermal regime of Himalayan mountain glaciers"

_The Cryosphere, 2019_

## Referee Comment (RC1) · Anonymous Referee #1 · 9 Sep 2019

**General comments**

I think this is an interesting paper that would be interest to readers of The Cryosphere. It does a good job at making the case that mountain glacier thermal structure is plausibly influenced by crevasse distribution and deep meltwater percolation. The overall quality of the scientific work appears to be good, although I called out several questions and concerns about both the methods and the reproducibility below. I don't think any problems that I bring up are necessarily fundamental to the study and uncorrectable.

The figures are very nice. As a general suggestion, I would recommend choosing colourmaps that are likely to be better preserved in print form (i.e., the "jet" colourmap

is virtually useless in black and white).

Regarding presentation, I've made a few suggestions that I think might improve the comprehensibility of the paper. I also recommend a thorough language-proofing.

**Specific comments**

Much of the argument relies on the correlation between observed surface crevassing and radar scattering, similar to what has been previously linked to englacial temperate ice. An alternate hypothesis might be that surface crevassing causes the scattering the radar signal, which I think should be discussed. Is it possible to discount this based on the timing of the scatter in the radargrams (or by some other means)? If so, I think it would strengthen the paper's argument to do so.

I also found some of the description of the procedure used to model the firn thickness and enthalpy distribution a little hard to follow, and a less ambiguous format (such as pseudocode) would help. Some of the equations seem to use undefined or non-canonical values.

What work has been undertaken to demonstrate the numerical validity of the procedures described in sections 3.3 and 3.4? (It's possible that a reference could be provided for 3.3, if prior art exists.) I found section 3.4 particularly hard to follow. It might be nice to structure in the form of a list of assumptions that are being made to put the existing derivations in context. A diagram demonstrating the geometry of the problem would also help.

On a similar note, will any of the modelling code be made available, similar to the radar data? I feel that doing so would go a long way to improving the reproducibility/auditability of this paper and the included methods (and code is so easy to distribute now that it would be wonderful to do so).

**Technical corrections**

(grammatical suggestions annotated as "(gr)")
16: (gr) "In cold and arid climates"

18: How does GPR reveal temperate ice? Is it more correct to say that GPR suggests/implies temperate ice due to bed reflectivity measurements?

21: (gr) "Model experiments show"

23: (gr) "The time scale of thermal regime change"

25: (gr) "without the effect of the crevasses"

41: (gr) provide direct observations of the glaciers' thermal condition

42: (gr) gives only

48: s/localization/location

51: (gr) "two previous"?

56: no semi-colon

59: (gr) "to draw conclusions"?

65: (gr) capitalize "Valley"

67: (gr) "in the vicinity"

71: Can any additional information be provided about the type of antenna and transmitter used? How wide is the frequency band over which energy is produced? Is this a frequency domain or impulse-type system?

73: Could you clarify whether all reflectors were picked, or only the one thought to represent the glacier base?

82: What's the spatial resolution of this imagery and of the DEM produced?

86: Perhaps the model variogram parameters could be shared in an appendix?

95: (gr) "greater than what could be supported by ice deformation alone and thus

implies basal sliding"

103: I'm not sure that I understand "are calibrated ... by linear regression method" well enough that I could reproduce this independently given the data. Could you describe in greater detail (or if this is done elsewhere, provide a reference)?

eq1: do we constrain R as less than to equal to M, or is it possible for refreezing to exceed melting (either locally or over the entire domain)?

125: I don't think the description for $r_\text{s/m}$ is correct. As written, one would expect that it could never exceed 1, however in eq4 it's clear that it can (i.e. in the accumulation zone). Perhaps it would be better described as "the ratio of annual accumulation to melt"?

eq3: Doesn't the second case imply that if the annual accumulation is half the melt, we model the radiative melt factor as if the surface we ice-covered half the time? It's not obvious to me that this should be true.

eq5: Since this is an annual precipitation rate summed over 365 days, don't we need to divide by 365 somewhere? I also find the units given for dP/dz surprising - I would have expected it to be m w.e. / (a m) to match Pref.

144: Is this parameterized in any way? Coefficients or parameterizations used in the flow law should be described.

153: (gr) "well-constrained,"

eq8: It'd be nice to drop the parentheses around T to disambiguate it from function application (as it's used on the LHS and in one of the integration limits). Also, I'm not sure we've defined $T_m(P)$ yet.

171: can this assumption (crevasses go to the bedrock) be justified?

184: I think this section would benefit from a listing with pseudocode explaining the various steps here. From how I understand it, you're doing somthing like

1. start with initial temperature T 2. compute strain rate edot from T (KP2010, equation not given) 3. compute initial Qlat with eq10 4. compute H from the edot and Qlat with eq7 assuming dh/dt is zero 5. compute T from H with eq8 and goto 2

197: Does the grid move with the surface or is it fixed?

202: (gr)

209: Can we simplify eq12 by expressing df in aˆ-1 (to avoid the scaling parameter)?

215: In equation 14, should F be Fref?

216: The notation might be a bit muddled here - the function is parameterized by zf (undefined, I think?), but that doesn't appear on the RHS. A diagram might help.

225: I'm pretty sure the units don't work out here - c(t) is accumulation per day, but df.F is per second

226: Again, I think a more structured way of describing the algorithm, like a simplified code listing, would be helpful; the text feels too ambiguous.

234: "shifted our temperature forcing" - how and how much? (If this is described later, I had a hard time making the connection.)

236: I'm not quite sure at first what "reported to the bedrock topography" means; are you altering the bedrock by an amount equal to the free surface change? Later in the paper I gather that you also changed the bedrock significantly in places where there is ground truth, which I think deserves justification.

246: "reasonable accordance with the observation" - what exactly does this mean, and what are the criteria for "reasonable"? And do we even expect the observations to be similar to the steady state?

261: How do we know that scattering isn't due to the crevassing itself?

287: s/inexistent/nonexistent

325: (gr) not a complete sentence

339: I presume this means "no snow or firn over the surveyed part of the glacier," as there does seem to be a large part of the glacier above the ELA where we might expect firn?

340: What are the uncertainties in the surface DEM? Are they meaningful (i.e., they're used as an input to create the modelled bedrock topography, IIUC)?

346: It's interesting that these uncertainties are much smaller than the differences between the measured is thickness and the modelled ice thickness, and it's hard to believe that 20 m of horizontal uncertainty account for the rest. Any idea where the remaining difference could come from? Is it all from the assumed friction parameter?

356: s/delicate/difficult

359: I might be misinterpretting this statement, but I'm not sure I agree; the advection of heat should depend on whether motion is at the surface or distributed throughout the thickness, shouldn't it?

Figure 1: This should list the UTM zone in (b). Can we also add the glacier outline to (b) as in Figure 3?

Figure 2: Is it possible to demarcate the (approximate) extent of the surface crevasses? (the labels don't do a very good job of indicating how wide the crevassed area is)

Figure 3: The panes would be more comparable if they used the same colour scale (i.e. currently one is [0, 150] m and the other is [0, 200] m)

582: s/localization/locations

588: s/localization/location

Figure 8: difficult to distinguish between modelled background and observed dots In figs 8,9, would be helpful to label the columns (i.e. no percolation, with observed
crevasses, with modelled crevasses)

Figure 10: Again, the modelled vs measured points in the map are difficult to distinguish

---

## Referee Comment (RC2) · Martin Lüthi (Referee) · 27 Dec 2019

**Review Gilbert et al**

The manuscript *The influence of water percolation through crevasses on the thermal regime of Himalayan mountain glaciers* by Gilbert and others is a very nice study on the polythermal regime of mountain glaciers. The paper describes a good modeling study that can explain the observed interesting polythermal structure.

Nevertheless, a lot of small changes are needed to render the manuscript ready for publication. Obviously, we as non-native English "users" have a big disadvantage, and our formulations might be not idiomatic. Despite this, care should be taken to at least consistently use "the", (the apparently random) capitalization rules, proper words ("localization" instead of "position" or "area", English – if somewhat similar – is not some kind of French), etc.

Once the many below concerns have been addressed (and many more which I did not all mention, since this is more time consuming than just rewriting the text), the paper will be ready for publication.

Sincerely, Martin Lüthi

**General comments**

Overall, this is a very nice paper on an important topic. The modeling study is nicely crafted, comes to meaningful results, and elucidates some very relevant processes in polythermal glaciers. What I really liked is

- good and comprehensive Introduction,

- very good and thorough approach to the problem at hand,

- very nice modeling study to explain noteworthy features in a unique data set,

- good, meaningful Discussion of the important processes and their general significance.

This being said, the paper unfortunately is presented with many small warts that should be cured before it is ready for publication.

It took me unusually long to write this review, especially since I commented on many small things that should have been improved by the proof-readers before submitting the manuscript. Using colons (:) before equations, citing equations without parentheses around equation numbers, using $\times$ for multiplication in formulas etc. are things I have never seen in any of the usual journals. Please adapt to the conventions of the journal. Also, the paper would certainly profit from streamlining by a native speaker (I only indicated a few (many!) small issues).

In the Methods section care should be taken to clearly describe the algorithms used, especially these rather ad-hoc rules that describe meltwater percolation. Especially section 3.3 is very opaque, and would profit from a flow diagram or a formula describing the iterations (I think).

Also the algorithm in Section 3.4.2 is not easily understood, and details on the procedure are missing in some steps (e.g. Step 2).

Some formulas (e.g. Eq. 14) appear to contain errors (outlined below).

Some figures, and many figure captions should be adapted and improved.

My feeling is, that the main results could be explained with less figures, and some of them could be moved to an appendix / supporting material section.

The bibliography appears complete with DOIs (although given as URL). ISBN/ISSN should be given for the cited books.

**Specific comments**

"the small number of boreholes gives …"

"extrapolated". The whole sentence is somewhat awkward, better reformulate and split in two.

"Scattering of electromagnetic waves …"

start new sentence after "GPR data".

also Ryser et al. (2013) nicely showed the relation between ice temperature and scattering.

"rare" instead of "rearer"

or $168 \, \mathrm{m} \, \mu \mathrm{s}^{-1}$

better "imagery"

"the Kriging algoritm" (give reference).

and 94: "stake" (singular)

"support" seems wrong here, better use "can be explained". And then, one wonders which ice flow parameters and stress regime have been used to arrive at this conclusion.

Since ice temperature is the main focus of this study, one is curious about the measurement process. Were the holes drilled mechanically or thermally? Was temperature measured once, or logged? Type of sensors and measurement equipment would also merit description.

"as input data" (leave away "an")

leave away "method"

"aims at identifying"

"observed": better say "deduced from"

no comma after "which"

leave away the colon (here, and before all equations). TC does not use this style.

only give units once (they have to be the same anyway).

this contradicts line 111. Only say you used and improved model there.

How does short-wave radiation affect ice dynamics (leave away)?

from where do you know the $f_{rad}$ values?

what value do all these factors have? Are they taken from literature, or determined from local measurements?

Using $Q$ for a source term is unfortunate notation, since often $Q$ denotes fluxes. Indeed, it is called a flux on line 174.

$\sigma$ and $\varepsilon$ should also be written bold in the text

"constrained"

replace "defined" by "written in terms of"

A dot is missing in the number $3 \cdot 10^3$

Here it would be interesting to say what the vertical resolution in meters is.

What kinds of elements have been used? Geometry and approximation order should be mentioned here.

Why and how is water percolating to the bed? Ice (even temperate ice) is pretty impermeable.

"neglected"? Not clear what you want to say with this. Neglected from what?

Why a heat flux, and in which direction? This should be a source term in Equation (7).

How is the amount of refreezing water determined?

Top to bottom of what? Of the glacier or of the layer?

"fusion of water" is, I think, simply $T_m$.

"the water can access"

I don't think $Q_{lat}$ is a flux. It's a source term.

This whole description of the algorithm is somewhat opaque. Please consider a flow diagram, or a clear description of what happens.

"steady state": what are you doing here? Do you mean you do a fixed-point iteration until convergence? Are the steps iterative steps (for the solution at one grid point), or time steps? I doubt that you calculate a steady state of the whole glacier here.

"distribute": I think you use the 1-d model at every grid point, independent of all other grid points? If so, please say that.

"for the 3D model"

"It provides a high …"

how sensitive is the temperature profile to the choice of the Gaussian standard deviation?

"reaches"

why not just using a commensurate value $d$ with time unit in years$^{-1}$, then $t_{yr}$ could be left away.

"Combining Equations (13) and (14) gives" with parentheses around equations, "Equations" capitalized, and the sentence without colons (like any other article in TC).

The time step should be named $\Delta t$ (not the infinitesimal $dt$), and already named in the text.

why is "Topography" capitalized. Even if English has no meaningful rules, titles should be capitalized consistently.

"known" instead of "resolved"

"the" instead of "our"

leave away parentheses around $\beta$.

Was this "best" determined with an optimization?

What is the meaning of these units?

What is done during the "reporting"? This is probably not the right expression, and something happens to update the bed topography.

"constraining with"?

"observations".

"permits" instead of "allows"

"were performed" (also the past/present tense should be used consistently)

I'm not sure whether a comparison to other glaciers would rather belong to the Discussion.

Why not simply "that the occurrence of temperate ice …"

Reference needed for ERA.

"equilibrium line altitude" (lowercase)

"provides" sounds wrong, why not "the model is in good agreement with …"

"in areas without radar measurements"

weird sentence, please correct

"in equilibrium"

"the climate change" is pretty meaningless. Probably you mean "surface warming" or similar? Also line 317.

please call this consistently "ERA-interim". The time series should also be shown, maybe in Figure 4c.

"crevasses"

"linearly increasing temperature", the trend is not increasing.

Do you mean polythermal glaciers in general. Then the "the" should be discarded.

this sentence is incomplete.

weird sentence weird sentence

"values". Better rephrase

"calibrated on data" is not proper English (IMHO)

A GPS (Garmin) should be accurate to 5-10 meters, so what is the problem here?

Considering how much effort it is doing these measurements, why don't you use some cheap real-time corrected GPS, such as the Emlid Reach?

"the friction …"

complicated sentence, rephrase

Why is this "mass flux conservation" special? It is part of the solution of the Stokes equations.

"coming from": better "derived from surface melt"

"the Greenland …"

"the thermal regime"

not really "observed", but "inferred for"

"position" instead of "localization"

"warming" instead of "climate change" (which is a generic catch phrase without any particular meaning for this mountain area – there could also be local cooling)

"combined"

"reveals"

"crevassed areas" or "crevasse fields"

"facilitates/permits/enables" instead of "allows"

"affects"

"the thermal…"

"surface temperature increase" or "surface warming" (again, climate change is unspecific)

Eq 5 Why are you using % and (1/100) here? Its easier to understand if you just use the ratios.

Eq 12 write equations without "×" (also in many others)

Eq 14  the integration variable should be $dz$, not $dz_f$. Maybe the upper integration limit should be $z_f$. This should be written carefully!

Eq 16  Do not use $\times$ in any of these formulas, they become unreadable.

Fig 1  (a) The black line around the glacier is barely visible, use orange?

(b) Could the location of the thermistor chain be indicated with something more distinctive, e.g. a red diamond.

Fig 2  The caption should also mention what we see, i.e. cold and temperate zones (with and without reflections). An approximate distance and depth scale should also be given. Microseconds could be used instead of 1000s of nanoseconds.

Fig 3a  The black dashed line is really hard to see. This line should also be explained in the caption. It is also quite unfortunate to use different depth scales in the panels.

Fig 4  caption (585) please use correct English for plural: "stake positions" / "stake measurements"

Fig 5  caption (588) "at the three stake positions" (we already know that they are different)

Fig 5  caption (589) What does "steady" mean in this context of an oscillating forcing? Is this a limiting cycle (stationary periodic response at depth)?

Fig 6  caption: "localization": better say "positions"

Fig 7  panels (a) and (c): the half-transparent colors of the plot are different from those of the color bar. Use the full colors (alpha=1). This also applies to other figures (9-12).

Fig 7  caption: "localization": better say "crevassed areas"

Fig 8  upper panels: "temperate" in lower case on both axes

Fig 8  lower panels: the dots are very difficult to see. Maybe a white border around them would help?

Fig 8  a minor detail, but why are panel letters not placed as in Figure 6, 7 and 9?

Fig 9  Could the temperate basal areas be shown by a red color? In this color scheme the changes are too gradual for this very important switch. Are the temperatures here absolute, or relative to the pressure melting point (which is the only meaningful quantity to show here)?

Fig 10  caption (609): "the ablation zone"

Fig 11  caption (616): these are longitudinal sections (along flow line), not cross sections (across ice flow). Also correct this in all captions and the main text.

Tab 1  proper notation uses a central dot, not a cross ($5 \cdot 10^7$, not $5 \times 10^7$)

Tab 2  "Precipitation Lapse rate": consistent capitalization! Also the units are weird, why not just $m^{-1}$ (although I think this should be $a^{-1}$, so this is completely wrong).

Tab 2  Are the radiative rates per square meter? So the units are wrong.

**References**

Ryser, C., Lüthi, M., Blindow, N., Suckro, S., Funk, M., and Bauder, A. (2013). Cold ice in the ablation zone: its relation to glacier hydrology and ice water content. *Journal of Geophysical Research*, 118(F02006):693–705.

---

## Author Comment (AC1) · 2 Mar 2020

**The influence of water percolation through crevasses on the thermal regime of Himalayan mountain glaciers**

Adrien Gilbert et al.

**Response to reviewers' comments**

We would like to sincerely thank the referees for their careful feedback on our study that certainly helped to improve its presentation. We believe we can sufficiently respond to all comments made and improved the manuscript accordingly. The response to both reviewers is reported bellow.

Reviewer comments in normal font.
Response in *italic blue* font

**Referee #1**

**General comments**

I think this is an interesting paper that would be interest to readers of The Cryosphere. It does a good job at making the case that mountain glacier thermal structure is plausibly influenced by crevasse distribution and deep meltwater percolation. The overall quality of the scientific work appears to be good, although I called out several questions and concerns about both the methods and the reproducibility below. I don't think any problems that I bring up are necessarily fundamental to the study and uncorrectable.

*Thank you for the positive feedback and constructive comments. We have revised the manuscript accordingly.*

The figures are very nice. As a general suggestion, I would recommend choosing colourmaps that are likely to be better preserved in print form (i.e., the "jet" colourmap is virtually useless in black and white).

*We tried to improve our colourmaps as much as possible in that sense.*

Regarding presentation, I've made a few suggestions that I think might improve the comprehensibility of the paper. I also recommend a thorough language-proofing.

*The entire manuscript has been language edited.*

**Specific comments**

Much of the argument relies on the correlation between observed surface crevassing and radar scattering, similar to what has been previously linked to englacial temperate ice. An alternate hypothesis might be that surface crevassing causes the scattering the radar signal, which I think should be discussed. Is it possible to discount this based on the timing of the

scatter in the radargrams (or by some other means)? If so, I think it would strengthen the paper's argument to do so.

*We agree that it would be difficult to conclude about the presence of temperate ice within the crevassed areas where strong reflectors already appear near the surface due to crevassing. However, the way that the scattering is then advected by ice flow shows that the crevassed areas behave more as a source term of the scattering. It is unlikely that the advected scattering is due to advected crevassed ice since the scatter occurs in the deeper part of the glacier where crevasses are not expected to persist. It would be rather more compatible with the formation of temperate ice which is then advected by ice flow. Similar scattering due to temperate ice has also been confirmed by comparison of the radar profiles with borehole temperature measurements (e.g. Wilson et al. 2013). The hypothesis is confirmed by the modeling part of the paper. A short discussion has been added in the revised manuscript (lines 275-279).*

I also found some of the description of the procedure used to model the firn thickness and enthalpy distribution a little hard to follow, and a less ambiguous format (such as pseudocode) would help. Some of the equations seem to use undefined or non-canonical values.

*We improved this section according to your comments (in technical correction) and those from reviewer 2. We also checked that all parameters and variables are well defined in all the equations.*

What work has been undertaken to demonstrate the numerical validity of the procedures described in sections 3.3 and 3.4? (It's possible that a reference could be pro-vided for 3.3, if prior art exists.) I found section 3.4 particularly hard to follow. It might be nice to structure in the form of a list of assumptions that are being made to put the existing derivations in context. A diagram demonstrating the geometry of the problem would also help.

*The section 3.4.1. is based on a 1D model which is published (Gilbert et al., 2014b) as mentioned in the manuscript. However, for this study, we had to model the firn and snow thickness to determine the density profile. The approach we propose here has not been published or validated before but we consider our study as a validation of this method. Same thing for the section 3.3, we propose a simple method to take into account water percolation in crevasses and the study is a validation of this approach.*

*The method presented in section 3.4.2 has already been used and validated in (Gilbert al., 2018), we added the reference.*

*According to the specific comments of the two reviewers, we have reorganized and partially rewritten these two sections, which we now believe, are easier to follow.*

On a similar note, will any of the modelling code be made available, similar to the radar data? I feel that doing so would go a long way to improving the reproducibility/auditability of this paper and the included methods (and code is so easy to distribute now that it would be wonderful to do so).

*The modeling code (section 3.3) will be integrated in the Elmer/Ice package which is freely available. It will include an example based on the Rika Samba simulation presented in this paper. The rest of the modeling uses solvers already available in Elmer/Ice. See http://elmerfem.org/elmerice/wiki/doku.php. We added this information in the data availability section.*

**Technical corrections**

(grammatical suggestions annotated as "(gr)")

16: (gr) "In cold and arid climates"
*Done*

18: How does GPR reveal temperate ice? Is it more correct to say that GPR suggests/implies temperate ice due to bed reflectivity measurements?
*We modified the sentence: "However, scattering in Ground Penetrating Radar (GPR) measurements on Rikha Samba Glacier in the Nepal Himalaya suggests a large amount of temperate ice that seems to be influenced by the presence of crevassed areas" (lines 17-18).*

21: (gr) "Model experiments show"
*Done*

23: (gr) "The time scale of thermal regime change"
*Done*

25: (gr) "without the effect of the crevasses"
*Done*

41: (gr) provide direct observations of the glaciers' thermal condition
*Done*

42: (gr) gives only
*Done*

48: s/localization/location
*Done*

51: (gr) "two previous"?
*Done*

56: no semi-colon
*Done*

59: (gr) "to draw conclusions"?
*Done*

65: (gr) capitalize "Valley"

*Done*

67: (gr) "in the vicinity"
*Done*

71: Can any additional information be provided about the type of antenna and transmitter used? How wide is the frequency band over which energy is produced? Is this a frequency domain or impulse-type system?
*Description of the antenna and transmitter have been added to the text also clarifying that it is impulse-type system with a frequency band of 15-45 MHz (lines 71-72).*

73: Could you clarify whether all reflectors were picked, or only the one thought to represent the glacier base?
*The reflectors were picked from the assumed glacier bed when possible as well as from the interface between cold and temperate ice identified from the signal scattering to quantify the thickness of cold and temperate ice. Clarification added in the text (lines 75-76).*

82: What's the spatial resolution of this imagery and of the DEM produced?
*This is now specified in the revised manuscript (lines 85).*

86: Perhaps the model variogram parameters could be shared in an appendix?
*The model variogram sill and range has been added in the text (lines 90).*

95: (gr) "greater than what could be supported by ice deformation alone and thus implies basal sliding"
*The sentence has been corrected*

103: I'm not sure that I understand "are calibrated ... by linear regression method" well enough that I could reproduce this independently given the data. Could you describe in greater detail (or if this is done elsewhere, provide a reference)?
*We now provide more detail here (lines 109-111).*

eq1: do we constrain R as less than to equal to M, or is it possible for refreezing to exceed melting (either locally or over the entire domain)?
*R is locally less than to equal to M (it is defined by Eq. 6).*

125: I don't think the description for $r_{s/m}$ is correct. As written, one would expect that it could never exceed 1, however in eq4 it's clear that it can (i.e. in the accumulation zone). Perhaps it would be better described as "the ratio of annual accumulation to melt"?
*Ok, we modified the sentence.*

eq3: Doesn't the second case imply that if the annual accumulation is half the melt, we model the radiative melt factor as if the surface we ice-covered half the time? It's not obvious to me that this should be true.
*Yes, this the principle of this parametrized approach. Since the parameters $f_{rad}^{ice}$ and $f_{rad}^{snow}$ are calibrated to fit the mass balance data, their values may compensate the uncertainty linked to the approximation of $r_{s/m}$ made here. This approach has been used in Gilbert et al. (2016)*

*where it provided a robust estimate of the surface mass balance despite of the crude approximation made to calculate $r_{s/m}$.*

eq5: Since this is an annual precipitation rate summed over 365 days, don't we need to divide by 365 somewhere?
*Yes, this has been modified in the revised manuscript.*
I also find the units given for dP/dz surprising - I would have expected it to be m w.e. / (a m) to match Pref.
*The reference precipitation $P_{ref}$ is multiplied by a factor depending of elevation which is expressed by (1+(z - $z_{ref}$)*dP/dz). To keep homogenous units (z - $z_{ref}$)*dP/dz should be without unit so dP/dz is in $m^{-1}$ . In the previous version of the manuscript we made a mistake in this factor, this has been corrected (line 137).*

144: Is this parameterized in any way? Coefficients or parameterizations used in the flow law should be described.
*This is not parametrized, the Stokes equation are fully solved. It has been clarified.*

153: (gr) "well-constrained,"
*Done*

eq8: It'd be nice to drop the parentheses around T to disambiguate it from function application (as it's used on the LHS and in one of the integration limits). Also, I'm not sure we've defined $T\_m(P)$ yet.
*Done. $T_m$ is defined in the sentence following the equation.*

171: can this assumption (crevasses go to the bedrock) be justified?
*Our assumption is not that crevasses go to the bedrock but that liquid water is able to percolate through ice in the crevassed areas via cracks opening under water pressure. Indeed, it has been shown that meltwater is able to penetrate up to 400m of cold ice in Greenland (Lüthi et al., 2015). This is also theoretically confirmed by Van Der Veen (2007) where the author shows that crevasses subjected to inflow of water will continue to propagate downward until the bedrock with the propagation speed controlled primarily by the rate of water injection. Those references have been added in the revised manuscript (line 178).*

*References:*

*Lüthi, M. P., Ryser, C., Andrews, L. C., Catania, G. A., Funk, M., Hawley, R. L., Hoffman, M. J. and Neumann, T. A.: Heat sources within the Greenland Ice Sheet: dissipation, temperate paleo-firn and cryo-hydrologic warming, The Cryosphere, 9(1), 245–253, doi:10.5194/tc-9-245-2015, 2015.*

*Van der Veen, C. J.: Fracture propagation as means of rapidly transferring surface meltwater to the base of glaciers, Geophys. Res. Lett., 34(1), L01501, doi:10.1029/2006GL028385, 2007.*

184: I think this section would benefit from a listing with pseudocode explaining the various steps here. From how I understand it, you're doing something like:

1. start with initial temperature T
2. compute strain rate edot from T (KP2010, equation not given)
3. compute initial $Q_{lat}$ with eq10
4. compute H from the edot and $Q_{lat}$ with eq7 assuming dh/dt is zero
5. compute T from H with Eq. 8 and goto 2

*We now provide a list of step to clarify the procedure used here to estimate $Q_{lat}$ and the steady state temperature field (lines 238-242).*

197: Does the grid move with the surface or is it fixed?
*The grid moves with the surface and the variables are interpolated from the old grid to the new grid at each time step. This is now specified in the revised manuscript.*

202: (gr)
*Done.*

209: Can we simplify eq12 by expressing df in aˆ-1 (to avoid the scaling parameter)?
*Ok done.*

215: In equation 14, should F be $F_{ref}$?
*No, F is the firn thickness at a given time whereas $F_{ref}$ is the initial firn thickness.*

216: The notation might be a bit muddled here - the function is parameterized by zf (undefined, I think?), but that doesn't appear on the RHS. A diagram might help.
*We re-organize the description of the computation of the density profile which is now clearer. We also checked that all the variables and parameters are well defined.*

225: I'm pretty sure the units don't work out here - c(t) is accumulation per day, but df.F is per second
*The equations are now homogeneous.*

226: Again, I think a more structured way of describing the algorithm, like a simplified code listing, would be helpful; the text feels too ambiguous.
*We re-organize the text in a way that the procedure we have done here is not ambiguous anymore.*

234: "shifted our temperature forcing" - how and how much? (If this is described later,I had a hard time making the connection.)
*This is described in the result section 4.2: "Balanced conditions for the 2014 geometry are reached for a climate that is 0.7°C colder than the 1980-2016 climate with an Equilibrium Line Altitude (ELA) of 5770 m a.s.l. (1980-2016 ELA is 5880 m a.s.l.; Fujita and Nuimura, 2011). This provides a mean surface mass balance and a melting rate to force the steady state glacier simulation". We now provide more details in this section 3.4.2. (lines 250-251).*

236: I'm not quite sure at first what "reported to the bedrock topography" means; are you altering the bedrock by an amount equal to the free surface change? Later in the paper I gather

that you also changed the bedrock significantly in places where there is ground truth, which I think deserves justification.

*Yes, this is what we have done, it is now clarified. Our bedrock correction avoids major flow divergence in the velocity field that would result in strong vertical ice advection that lead to unrealistic steady state temperature field. Also radar measurements have their own uncertainties and cannot be considered as ground truth especially in temperate area where the bedrock reflection can be pretty weak and undetermined. We therefore favor a bedrock topography that satisfy mass conservation with the prescribed surface mass balance. Also Figure 6c shows that bedrock correction where radar measurement are existent rarely exceed 20 m apart of a really few exception (5 points of the 2010 measurements). We added a comment in the revised manuscript about this point (lines 306-307).*

246: "reasonable accordance with the observation" - what exactly does this mean, and what are the criteria for "reasonable"? And do we even expect the observations to be similar to the steady state?

*We mean here that, for the purpose of the study, which is a thermal regime study, our method allows a good approximation of the glacier dynamic and geometry. We delete this sentence here since the result section provides the quantitative comparison needed to evaluate how good the accordance is. This comment in our manuscript was unnecessary here.*

261: How do we know that scattering isn't due to the crevassing itself?

*See explanation in the General Comments. We added a short discussion in the revised manuscript (lines 275-279)*

287: s/inexistent/nonexistent
*Done*

325: (gr) not a complete sentence
*The sentence has been corrected.*

339: I presume this means "no snow or firn over the surveyed part of the glacier," as there does seem to be a large part of the glacier above the ELA where we might expect firn?

*The mean ELA over the period 1980-2016 is at about 5880 m (Figure 4b) (contrary to the steady state ELA at 5760 m (Figure 4a)). Over the last ten years the ELA was even above 5900 m which is in accordance with the fact that no firn were present over the surveyed part of the glacier in 2015.*

340: What are the uncertainties in the surface DEM? Are they meaningful (i.e., they're used as an input to create the modelled bedrock topography, IIUC)?

*Surface DEM was made from high resolution Pleiades images resulting in uncertainty bellow 1m. The surface DEM was acquired on November 7, 2014, 6 months before the radar measurement. With a mean thinning rate of 0.5 m w.eq. yr$^{-1}$, the uncertainty introduced by the temporal lag between the two measurements (radar and surface DEM) should be inferior to 50 cm making the uncertainty on the surface topography negligible in comparison to the one coming from ice thickness estimation. The bedrock topography uncertainties are therefore not significantly affected by the surface DEM uncertainties. We added a sentence in the revised manuscript (line 370).*

346: It's interesting that these uncertainties are much smaller than the differences between the measured is thickness and the modelled ice thickness, and it's hard to believe that 20 m of horizontal uncertainty account for the rest. Any idea where the remaining difference could come from? Is it all from the assumed friction parameter?

*These uncertainties are coming from the theoretical vertical resolution of the radar measurements. However, the ice/rock interface is manually picked on the radargram which can introduce extra uncertainty when the reflector is weak (especially in temperate area, see figure 2). And yes, as you pointed out, the modeled thickness is strongly linked to the friction parameter which is not known but also by the surface mass balance which is modeled and also introduces uncertainties in the reconstruction. We add a sentence in the revised manuscript (line 367).*

356: s/delicate/difficult
*Done*

359: I might be misinterpreting this statement, but I'm not sure I agree; the advection of heat should depend on whether motion is at the surface or distributed throughout the thickness, shouldn't it?

*Yes, this sentence is not clear, the way that motion is distributed is important. We meant here that if the modeled surface velocities are in accordance with the measurements, it is likely that the advection processes are well represented because the way that motion is distributed is solved through the stokes equation that should accurately represent ice deformation in our 3D setup. We removed this sentence of the manuscript which led to confusion.*

Figure 1: This should list the UTM zone in (b). Can we also add the glacier outline to(b) as in Figure 3?
*Done*

Figure 2: Is it possible to demarcate the (approximate) extent of the surface crevasses? (the labels don't do a very good job of indicating how wide the crevassed area is)
*Done*

Figure 3: The panes would be more comparable if they used the same colour scale(i.e. currently one is [0, 150] m and the other is [0, 200] m)
*Done*

582: s/localization/locations
*Done*

588: s/localization/location
*Done*

Figure 8: difficult to distinguish between modelled background and observed dots In figs 8,9, would be helpful to label the columns (i.e. no percolation, with observed crevasses, with modelled crevasses)
*We improved this figure.*

Figure 10: Again, the modelled vs measured points in the map are difficult to distinguish
*We improved this figure.*

**Referee #2 (Martin Lüthi)**

The manuscript *The influence of water percolation through crevasses on the thermal regime of Himalayan mountain glaciers* by Gilbert and others is a very nice study on the poly-thermal regime of mountain glaciers. The paper describes a good modeling study that can explain the observed interesting polythermal structure.

Nevertheless, a lot of small changes are needed to render the manuscript ready for publication. Obviously, we as non-native English "users" have a big disadvantage, and our formulations might be not idiomatic. Despite this, care should be taken to at least consistently use "the", (the apparently random) capitalization rules, proper words ("localization" instead of "position" or "area", English – if somewhat similar – is not some kind of French), etc.

Once the many below concerns have been addressed (and many more which I did not all mention, since this is more time consuming than just rewriting the text), the paper will be ready for publication.

Sincerely,
Martin Lüthi

**General comments**

Overall, this is a very nice paper on an important topic. The modeling study is nicely crafted, comes to meaningful results, and elucidates some very relevant processes in poly-thermal glaciers. What I really liked is

• good and comprehensive Introduction,
• very good and thorough approach to the problem at hand,
• very nice modeling study to explain noteworthy features in a unique data set,
• good, meaningful Discussion of the important processes and their general significance.

This being said, the paper unfortunately is presented with many small warts that should be cured before it is ready for publication.

It took me unusually long to write this review, especially since I commented on many small things that should have been improved by the proof-readers before submitting the manuscript. Using colons (:) before equations, citing equations without parentheses around equation numbers, using × for multiplication in formulas etc. are things I have never seen in any of the usual journals. Please adapt to the conventions of the journal. Also, the paper would certainly profit from streamlining by a native speaker (I only indicated a few (many!) small issues).

*We are grateful for the careful job you have done in pointing out all those small issues. We have taken into account all of them and the manuscript has also been streamlined by a professional English proofreader.*

In the Methods section care should be taken to clearly describe the algorithms used, especially these rather ad-hoc rules that describe meltwater percolation. Especially section 3.3 is very opaque, and would profit from a flow diagram or a formula describing the iterations (I think). Also the algorithm in Section 3.4.2 is not easily understood, and details on the procedure are missing in some steps (e.g. Step 2). Some formulas (e.g. Eq. 14) appear to contain errors (outlined below).

*With the help of reviewer 1 as well, we improved the clarity of those sections by adding more details and restructured how the different steps are ordered.*

Some figures, and many figure captions should be adapted and improved.
*We modified and improved all the figure according to your comments.*

My feeling is, that the main results could be explained with less figures, and some of them could be moved to an appendix / supporting material section.

*We decided to keep all the figures as 12 figures still fits the format requirements is The Cryosphere and we prefer to have all the information in a single document.*

The bibliography appears complete with DOIs (although given as URL). ISBN/ISSN should be given for the cited books.

*ISBN/ISSN have been added.*

**Specific comments**

41 "the small number of boreholes gives …"
*Done*

43 "extrapolated". The whole sentence is somewhat awkward, better reformulate and split in two.
*Done*

45 "Scattering of electromagnetic waves …"
*Done*

46 start new sentence after "GPR data".
*Done*

47 also Ryser et al. (2013) nicely showed the relation between ice temperature and scattering.
*We added this reference.*

50 "rare" instead of "rearer"

*Done*

74 or168 mμs−1
*Done*

84 better "imagery"
*Done*

86 "the Kriging algoritm" (give reference).
*Done. We added a reference.*

92 and 94: "stake" (singular)
*Done*

95 "support" seems wrong here, better use "can be explained". And then, one wonders which ice flow parameters and stress regime have been used to arrive at this conclusion.
*We made the change in the revised manuscript. This statement comes from the cited reference (Fujita et al., 2001) where the authors used measured ice thickness and slope to estimate surface velocities using realistic flow rate factor in the shallow ice approximation (slab). Comparison with observed surface velocities that are much higher led to the conclusion that sliding occurs under Rikha Samba Glacier. We added details in the revised version of the manuscript (lines 99-101).*

96 Since ice temperature is the main focus of this study, one is curious about the measurement process. Were the holes drilled mechanically or thermally? Was temperature measured once, or logged? Type of sensors and measurement equipment would also merit description.
*The details of the ice temperature measurements are added to the text. The ice temperatures were logged with a thermistor chain manufactured by Stump Bohr AG, Switzerland, over one-year time period.*

100 "as input data" (leave away "an")
*Done*

104 leave away "method"
*Done*

106 "aims at identifying"
*Done*

106 "observed": better say "deduced from"
*Done*

107 no comma after "which"
*Done*

112 leave away the colon (here, and before all equations). TC does not use this style.
*Done. We removed it before all equations*

115 only give units once (they have to be the same anyway).
*Ok done*

117 this contradicts line 111. Only say you used and improved model there.
*Ok done*

118 How does short-wave radiation affect ice dynamics (leave away)?
*We removed this sentence. It was not necessary here.*

125 from where do you know the frad values?
*Those parameters are used as tuning parameters and are calibrated to match the measurements. This is detailed in the result section and in Table 2.*

142 what value do all these factors have? Are they taken from literature, or determined from local measurements?
*This is also detailed in Table 2. We added a sentence to refer to the result section about this.*

151 Using Q for a source term is unfortunate notation, since often Q denotes fluxes. Indeed, it is called a flux on line 174.
*Using Q for the volumetric flux coming from latent heat release (often called "heating rate") is a common notation used in most literature. We kept the notation but now refers to heating rate instead of source term to be consistent with the literature refereeing to this term.*

152 σ and ε should also be written bold in the text
*Done*

153 "constrained"
*Done*

155 replace "defined" by "written in terms of"
*Done*

159 A dot is missing in the number $3 \cdot 10^3$
*The number seems to be correct at this line.*

162 Here it would be interesting to say what the vertical resolution in meters is.
*We now specify the vertical resolution here (~10 m).*

162 What kinds of elements have been used? Geometry and approximation order should be mentioned here.
*The elements we used are now described. "The mesh is constructed from 2D triangular linear element extruded in the vertical direction. It gives a mesh made of triangular prism unstructured in the horizontal direction and structured in the vertical direction." (line 169-170).*

171 Why and how is water percolating to the bed? Ice (even temperate ice) is pretty impermeable.

*The assumption that surface crevasses can initiate meltwater routing down to the bedrock is supported by several papers in the literature. Moreover, this also what is suggested by the radar measurement showing the development of temperate ice from the surface down to the bedrock in crevasse fields. We added a paragraph describing how this assumption can be supported by previous literature and by our measurements (line 177-178).*

173 "neglected"? Not clear what you want to say with this. Neglected from what?
*We meant that surface meltwater coming from outside of the crevassed area through surface runoff is neglected in the amount of water available for refreezing in crevassed area. The revised manuscript has been clarified (line 180).*

174 Why a heat flux, and in which direction? This should be a source term in Equation(7).
*We now call it heating rate, yes it is a source term.*

175 How is the amount of refreezing water determined?
*This is described in the following paragraph. The annual surface melting is distributed vertically as available latent heat from refreezing as long as the ice temperature is cold.*

179 Top to bottom of what? Of the glacier or of the layer?
*Of the glacier. We added it in the revised manuscript.*

180 "fusion of water" is, I think, simply Tm.
*Yes, we modified the text here introducing the enthalpy of fusion instead.*

180 "the water can access"
*Done*

184 I don't think Qlat is a flux. It's a source term.
*Yes, we now call it the heating rate.*

185 This whole description of the algorithm is somewhat opaque. Please consider a flow diagram, or a clear description of what happens.
*This was also pointed out by reviewer 1. We now clearly specify the different steps to make the description clearer. We also moved this part in the section 3.4 about steady state strategy. (line 239-244).*

186 "steady state": what are you doing here? Do you mean you do a fixed-point iteration until convergence? Are the steps iterative steps (for the solution at one grid point), or time steps? I doubt that you calculate a steady state of the whole glacier here.
*We calculate a steady state for the whole glacier here. We clarified the procedure in the revised manuscript (line 239-241).*

191 "distribute": I think you use the 1-d model at every grid point, independent of all other grid points? If so, please say that.
*Yes, we now specify this (line 200-201).*

193 "for the 3D model"

*Done*

194 "It provides a high ..."
*Done*

200 how sensitive is the temperature profile to the choice of the Gaussian standard deviation?
*The temperature profile is pretty sensitive to this choice since higher standard deviation lead to greater positive degree-day and therefore more melting. However, this standard deviation is well constrained by the fact we impose than the mean temperature seasonal cycle conserves the amount of positive degree day and therefore the amount of modeled melting. We added a sentence about this (line 210-211).*

203 "reaches"
*Done*

209 why not just using a commensurate valued with time unit in years−1, then tyr could be left away.
*Yes, we changed the units here.*
217 "Combining Equations (13) and (14) gives" with parentheses around equations, "Equations" capitalized, and the sentence without colons (like any other article in TC).
*This has been corrected.*

222 The time step should be named Δt (not the infinitesimal dt), and already named in the text.
*Done*

227 why is "Topography" capitalized. Even if English has no meaningful rules, titles should be capitalized consistently.
*We tried to be now consistent with capitalized words in the revised manuscript.*

229 "known" instead of "resolved"
*Done*

233 "the" instead of "our"
*Done*

234 leave away parentheses around $\beta$.
*Done*

235 Was this "best" determined with an optimization?
*This best was manually estimated to minimize the difference between measurements and model. Indeed, we assumed a uniform coefficient in this first step. This is now specified.*

235 What is the meaning of these units?
*This is the value of the uniform coefficient $\beta$ mentioned in this first step. We moved the value sooner in the sentence to make this clear.*

236 What is done during the "reporting"? This is probably not the right expression, and something happens to update the bed topography.
*This now clarified as suggested by the reviewer 1 as well (line 254).*

239 "constraining with"?
*Done*

246 "observations".
*This sentence has been removed (see reviewer 1 comments).*

246 "permits" instead of "allows"
*This sentence has been removed (see reviewer 1 comments).*

249 "were performed" (also the past/present tense should be used consistently)
*Done*

256 I'm not sure whether a comparison to other glaciers would rather belong to the Discussion.
*We moved this comparison to the discussion (line XX).*

260 Why not simply "that the occurrence of temperate ice …"
*Yes, we modified.*

264 Reference needed for ERA.
*Done*.

269 "equilibrium line altitude" (lowercase)
*Done*

271 "provides" sounds wrong, why not "the model is in good agreement with …"
*Yes, we corrected this.*

287 "in areas without radar measurements"
*Done*

303 weird sentence, please correct
*Done*

308 "in equilibrium"
*Done*

310 "the climate change" is pretty meaningless. Probably you mean "surface warming" or similar? Also line 317.
*Yes, we modified this.*

312 please call this consistently "ERA-interim". The time series should also be shown ,maybe in Figure 4c.

*Ok. We now added the annual temperature from ERA-interim in Figure 4.*

320 "crevasses"
*Done*

321 "linearly increasing temperature", the trend is not increasing.
*Done*

325 Do you mean polythermal glaciers in general. Then the "the" should be discarded.
*Yes, this sentence has been modified according to reviewer 1 comment.*

326 this sentence is incomplete.
*This sentence has been corrected.*

331 weird sentence
*We improved this sentence*

336 weird sentence
*We improved this sentence*

337 "values". Better rephrase
*Ok we changed "mean values" by "average"*

338 "calibrated on data" is not proper English (IMHO)
*We corrected the sentence.*

340 A GPS (Garmin) should be accurate to 5-10 meters, so what is the problem here?
*Even if the horizontal accuracy of the GPS is within 5-10 m, the accuracy of the positioning decreases when moving on the steep slopes in the mountainous terrain due to e.g. a weaker satellite signal, unevenly distributed satellite coverage or the GPR antenna not being completely aligned because of the rough terrain.*

344 Considering how much effort it is doing these measurements, why don't you use some cheap real-time corrected GPS, such as the Emlid Reach?
*In this paper, we used rental equipment with the GPS provided with the GPR system. Other, more accurate methods will be definitely worth using when possible in future.*

352 "the friction ..."
*Done*

355 complicated sentence, rephrase
*Done*

355 Why is this "mass flux conservation" special? It is part of the solution of the Stokes equations.
*Yes, the mass conservation is always satisfied when solving the Stokes equation but can lead to strong surface elevation change if the flux divergence significantly differs from surface mass*

*balance. We meant here that geometry is in equilibrium with surface mass balance. We clarified this in the revised manuscript (lines 378-379).*

362 "coming from": better "derived from surface melt"
*Done*

368 "the Greenland …"
*Done*

370 "the thermal regime"
*Done*

372 not really "observed", but "inferred for"
*Done*

384 "position" instead of "localization"
*Done*

385 "warming" instead of "climate change" (which is a generic catch phrase without any particular meaning for this mountain area – there could also be local cooling)
*Done*

392 "combined"
*Done*
393 "reveals"
*Done*

394 "crevassed areas" or "crevasse fields"
*Done*

395 "facilitates/permits/enables" instead of "allows"
*Done*

396 "affects"
*Done*

397 "the thermal…"
*Done*

398 "surface temperature increase" or "surface warming" (again, climate change is un-specific)
*Done*

Eq 5 Why are you using % and (1/100) here? Its easier to understand if you just use the ratios.
*Yes we changed this.*

Eq 12 write equations without "×" (also in many others)

*We removed the "x" in all the equations.*

Eq 14 the integration variable should be dz, not dzf. Maybe the upper integration limit should be zf. This should be written carefully!
*We corrected the equation.*

Eq 16 Do not use × in any of these formulas, they become unreadable.
*We removed the "x" in all the equations.*

Fig 1 (a) The black line around the glacier is barely visible, use orange?(b) Could the location of the thermistor chain be indicated with something more distinctive, e.g. a red diamond.
*We modified the figure accordingly.*

Fig 2 The caption should also mention what we see, i.e. cold and temperate zones (with and without reflections). An approximate distance and depth scale should also be given. Microseconds could be used instead of 1000s of nanoseconds.
*Done*

Fig 3a The black dashed line is really hard to see. This line should also be explained in the caption. It is also quite unfortunate to use different depth scales in the panels.
*We now use the same scale and change the color of the dashed line.*

Fig 4 caption (585) please use correct English for plural: "stake positions" / "stake measurements"
*Done*

Fig 5 caption (588) "at the three stake positions" (we already know that they are different)
*Done*

Fig 5 caption (589) What does "steady" mean in this context of an oscillating forcing? Is this a limiting cycle (stationary periodic response at depth)?
*Yes we modified the caption here.*

Fig 6 caption: "localization": better say "positions"
*Done*

Fig 7 panels (a) and (c): the half-transparent colors of the plot are different from those of the color bar. Use the full colors (alpha=1). This also applies to other figures (9-12).
*Done*

Fig 7 caption: "localization": better say "crevassed areas"
*Done*

Fig 8 upper panels: "temperate" in lower case on both axes
*Done*

Fig 8 lower panels: the dots are very difficult to see. Maybe a white border around them would help?
*We changed the colormap, it should be better now.*

Fig 8 a minor detail, but why are panel letters not placed as in Figure 6, 7 and 9?
*Done*

Fig 9 Could the temperate basal areas be shown by a red color? In this color scheme the changes are too gradual for this very important switch. Are the temperatures here absolute, or relative to the pressure melting point (which is the only meaningful quantity to show here)?
*Temperature are absolute here but, given the thickness of the glacier, the pressure melting point temperature does not differ significantly from 273.15K (reduced of maximum 0.15K). The difference would not be visible in our colormap. Temperate areas are highlighted by a dashed line that we now made more visible.*

Fig 10 caption (609): "the ablation zone"
*Done*

Fig 11 caption (616): these are longitudinal sections (along flow line), not cross sections (across ice flow). Also correct this in all captions and the main text.
*Done*

Tab 1 proper notation uses a central dot, not a cross (5·107, not5×107)
*Done*

Tab 2 "Precipitation Lapse rate": consistent capitalization! Also the units are weird, why not justm−1(although I think this should bea−1, so this is completely wrong).
*Done. The notation dP/dz is not adequate since we use a multiplicative factor, this why the unit is km-1. We changed the notation to avoid confusion.*

Tab 2 Are the radiative rates per square meter? So the units are wrong.
*$f_{rad}$ is multiplied by potential solar radiation ($W\ m^{-2}$) to obtain melting rate ($m\ w.eq.\ d^{-1}$). The unit ($m\ w.eq.\ d^{-1}\ W^{-1}\ m^2$) should be correct here.*

**References**
Ryser, C., Lüthi, M., Blindow, N., Suckro, S., Funk, M., and Bauder, A. (2013). Cold ice in the ablation zone: its relation to glacier hydrology and ice water content. Journal of Geophysical Research, 118(F02006):693–705.

---

## Author Comment (AC2) · 2 Mar 2020

The comment was uploaded in the form of a supplement:
https://www.the-cryosphere-discuss.net/tc-2019-172/tc-2019-172-AC2-
supplement.pdf

---

## Author Response (AR2)

**The influence of water percolation through crevasses on the thermal regime of a Himalayan mountain glacier**

Adrien Gilbert et al.

**Response to editor's comments**

Editor comments in normal font.
Response in *italic blue* font

[Title]
I agree that your approach can be applied to other Himalayan glaciers, however, this is not what you did in this paper. Therefore, to avoid unnecessary overstatements extending beyond the realm of the contents presented, I suggest to leave a singular form of a glacier (not "glaciers") as: "The influence of water percolation through crevasses on the thermal regime of a Himalayan mountain glacier".

*You are right, we originally choose this title because we wanted the paper to be process-oriented. There is no reason that this process is not occurring anywhere else so we choose a title that cover glaciers in general. It is however true that even if the process occurs, the extent of its influence will depend on the considered glacier. We changed the title according to your suggestion.*

[p.3, line 75]
"Continuous profiles were filtered and some of them migrated ..."
- there are so many ways to filter and introduce a phase shift (which I imagine behind your "migrated"), that I find it necessary to explain how exactly the filtering was done. Now the reader has no clue about this step. Please clarify.

*This has been clarified (lines 75-76)*

[p.7, line 174]
"We compare it with a threshold value $\sigma th$ (MPa) to identify where damage production occurs."
- I could not find where in text the chosen threshold value was shown. Was it 0.1 MPa? Why? Please, indicate it and support with a reference (there is a great variation in literature as you know).

*Yes, we forget to mention this value. We use a value of 0.05 MPa in order to better reproduce the observed crevassed areas. This value is within the recommended range found by Krug et al. (2014) ([0.01 0.2] MPa) who used similar approach to model calving events. We modify the revised manuscript (lines 304-305).*

[p.9, lines 238-243]
Please, put periods at the end of each sentence describing your iterative procedure steps.
*Done*

[p.10, line 250]

"we shifted the temperature forcing (1980-2016 corrected reanalysis) of 0.7C to obtain balanced mean conditions during the simulation period."
- I am not sure what is meant by "of" preposition here. Did you shifted "by -0.7C"? Why 0.7? Please, re-write in a more straightforward way.
*We corrected this sentence which now reads: "we shifted the temperature forcing (1980-2016 corrected reanalysis) by -0.7°C which allows to obtain balanced mean conditions during the simulation period."*

[p.12, line 333]
"we performed a transient simulation starting in 1975 from the modeled steady state (with crevasses location imposed from observations)"
- do you mean modern observations? Associated not with the past 1975 conditions, but from GPR and Google Earth in 2010-s? Please clarify.
*Yes, we meant modern observation. We clarified.*

[p.13, line 358]
should be able to adequately captured it.
->... capture
*Done*

[p. 23, Fig. 2 caption]
along the black dashed line in Fig. 3a.
-> I could not find such in Fig. 3a; should be red?
*Yes, we corrected the legend*

[p. 31, Fig. 10 caption]
"radar section next to the thermistor (black line in (e)) with the dashed line showing the modeled CTS."
- I could not follow which black line in (e) represents the radar section. Now (e) shows a longitudinal temperature profile 4 km long, where the upper black solid curve shows the surface and lower shows the base. Please re-check.
*We corrected this and now refer to the location shown in figure (b).*

- I do not think that your CTS abbreviation (cold-temperate transition surface) was defined anywhere. Please do so.
*Done*

[revised manuscript text omitted]
  shows the radar  measurement next to the thermistor (black circle in (b))  with the dashed line showing the modeled Cold-temperate Transition Surface (CTS). (b) Mean surface boundary condition over the period 1981-2016. (c) Modeled basal temperature. (d) Distribution of modeled and measured temperate ice thickness. (e) Modeled temperature along the radar longitudinal section presented in Fig. 2.**

[Figure]

Figure 11. Future evolution of Rikha Samba Glacier assuming a linear temperature increase of +1 °C between 2014 and 2100 (+1.7 °C in comparison with the steady state climatic condition). Upper panels represent basal temperature evolution. Lower panels are temperature evolution along the middle longitudinal section. Black dashed lines delimit the cold-temperate transition surface.

670

[Figure]

675 **Figure 12. Same as Fig. 11 but without water percolation in crevasses.**